# Paradoxical network excitation by glutamate release from VGluT3[+] GABAergic interneurons

**Kenneth A Pelkey[1†]\*, Daniela Calvigioni[1†], Calvin Fang[1], Geoffrey Vargish[1], Tyler Ekins[1], Kurt Auville[1], Jason C Wester[1], Mandy Lai[1], Connie Mackenzie-Gray Scott[1], Xiaoqing Yuan[1], Steven Hunt[1], Daniel Abebe[1], Qing Xu[2], Jordane Dimidschstein[3], Gordon Fishell[3,4], Ramesh Chittajallu[1], Chris J McBain[1]\***

[1]*Eunice Kennedy Shriver* National Institute of Child Health and Human Development, National Institutes of Health, Bethesda, United States; [2]Center for Genomics and Systems Biology, NYU, Abu-Dhabi, United Arab Emirates; [3]Stanley Center for Psychiatric Research, Broad Institute of MIT and Harvard, Cambridge, United States; [4]Department of Neurobiology, Blavatnik Institute, Harvard Medical School, Boston, United States

**\*For correspondence:**
pelkeyk2@mail.nih.gov (KAP);
mcbainc@mail.nih.gov (CJMB)

[†]These authors contributed equally to this work

**Competing interests:** The authors declare that no competing interests exist.

**Abstract** In violation of Dale's principle several neuronal subtypes utilize more than one classical neurotransmitter. Molecular identification of vesicular glutamate transporter three and cholecystokinin expressing cortical interneurons (CCK[+]VGluT3[+]INTs) has prompted speculation of GABA/glutamate corelease from these cells for almost two decades despite a lack of direct evidence. We unequivocally demonstrate CCK[+]VGluT3[+]INT-mediated GABA/glutamate cotransmission onto principal cells in adult mice using paired recording and optogenetic approaches. Although under normal conditions, GABAergic inhibition dominates CCK[+]VGluT3[+]INT signaling, glutamatergic signaling becomes predominant when glutamate decarboxylase (GAD) function is compromised. CCK[+]VGluT3[+]INTs exhibit surprising anatomical diversity comprising subsets of all known dendrite targeting CCK[+] interneurons in addition to the expected basket cells, and their extensive circuit innervation profoundly dampens circuit excitability under normal conditions. However, in contexts where the glutamatergic phenotype of CCK[+]VGluT3[+]INTs is amplified, they promote paradoxical network hyperexcitability which may be relevant to disorders involving GAD dysfunction such as schizophrenia or vitamin B6 deficiency.

## Introduction

With chemical transmission serving as the primary form of communication throughout the CNS, neurotransmitter identity fundamentally dictates neuronal circuit function. Indeed, at the first order of hierarchical classification systems neurons are functionally defined according to their classical neurotransmitter released during phasic synaptic communication. For example, almost all cortical neurons parse into glutamatergic excitatory neurons and gamma aminobutyric acid-ergic (GABAergic) inhibitory neurons. Despite scattered reports of coreleased neuromodulatory substances and peptides, up until the end of the last century it was widely accepted that individual neurons within the CNS release a single fast-acting neurotransmitter, an ideology formalized as Dale's principle (*Eccles, 1986*). However, the paradigm shifting observation of GABA and glycine corelease from spinal cord interneurons demanded reconsideration of this 'one neuron-one fast neurotransmitter' doctrine (*Jonas et al., 1998*).

Although GABA and glycine corelease could still be rationalized in the context that both substances exert similar rapid inhibitory influence on postsynaptic targets, subsequent reports demonstrated a rather astonishing prevalence of coreleased classic neurotransmitters from individual neurons capable of exerting disparate effects on postsynaptic targets (reviewed in *Granger et al., 2017*; *Hnasko and Edwards, 2012*; *Trudeau and El Mestikawy, 2018*). In particular, subpopulations of projection neurons from multiple subcortical monoaminergic systems, as well as local cholinergic interneurons, exhibit glutamate cotransmission with serotonin, dopamine, and acetylcholine (*Chuhma et al., 2014*; *Higley et al., 2011*; *Sengupta et al., 2017*; *Trudeau and El Mestikawy, 2018*; *Varga et al., 2009*). Remarkably, even the diametrically opposed cardinal excitatory and inhibitory transmitters glutamate and GABA serve as cotransmitters within several neuronal populations fundamentally challenging the basic tenet that fast synaptic signaling in the CNS occurs in a binary excitatory or inhibitory fashion (*Noh et al., 2010*; *Root et al., 2014*; *Root et al., 2018*; *Shabel et al., 2014*; *Yoo et al., 2016*).

The capacity for corelease directly relates to a neuron's abilities to intracellularly accumulate multiple transmitters, through uptake or synthesis, and load each species into synaptic vesicles (SVs) for regulated presynaptic membrane fusion. Canonically, GABA is synthesized from intracellular glutamate by one of two glutamate decarboxylases (GAD67 or GAD65 encoded by *Gad1* and *Gad2*, respectively), necessitating accumulation of two opposing classic neurotransmitters within most GABAergic neurons throughout the CNS. While ubiquitous expression of the vesicular GABA transporter (VGAT, encoded by *Slc32a1*) amongst central GABAergic neurons ensures efficient SV GABA filling, GABAergic cells typically lack any vesicular glutamate transporters (VGluTs). A conspicuous exception, are cortical cholecystokinin (CCK) and VGluT3 (encoded by *Slc17a8*) expressing interneurons (CCK$^+$VGluT3$^+$INTS), that are uniquely endowed with the enzymatic machinery necessary for both GABA and glutamate SV loading (*Somogyi et al., 2004*). Indeed, following the initial cloning of VGluT3, immunolocalization studies revealed coexpression with GADs and VGAT in perisomatic targeting GABAergic synapses from CCK$^+$ basket cells (CCK$^+$BCs; (*Fasano et al., 2017*; *Fremeau et al., 2002*; *Gras et al., 2002*; *Omiya et al., 2015*; *Rovira-Esteban et al., 2017*; *Schäfer et al., 2002*; *Somogyi et al., 2004*; *Stensrud et al., 2013*; *Stensrud et al., 2015*; *Takamori et al., 2002*). Moreover, recent unbiased single cell transcriptional profiling studies consistently reveal a discrete cohort of VGluT3$^+$, caudal ganglionic eminence-derived, CCK$^+$ rodent interneuron populations, which appears conserved in humans (*Harris et al., 2018*) (*Cck.Cxcl14.Slc17a8* and *Cck.Lypd1* clusters); (*Hodge et al., 2018*) (*L1-3PAX6SYT6* cluster); (*Tasic et al., 2018*) (*Sncg Slc17a8* cluster); (*Zeisel et al., 2015*) (*CCK Slc17a8 Cnr1* cluster)).

Based on the molecular/anatomical findings outlined above glutamate is posited to be coreleased with GABA from CCK$^+$VGluT3$^+$INTS for activation of presynaptic glutamate receptors to homo-/heterosynaptically regulate release or to activate postsynaptic ionotropic/metabotropic glutamate receptors (*Fasano et al., 2017*; *Omiya et al., 2015*; *Stensrud et al., 2015*). Such corelease is expected to impart unique computational properties to CCK$^+$VGluT3$^+$INTS differentiating them from CCKBCs lacking VGluT3 that are otherwise morphologically/electrophysiologically indistinguishable (*Bezaire and Soltesz, 2013*; *Oláh et al., 2019*; *Rovira-Esteban et al., 2017*; *Somogyi et al., 2004*). Indeed, CCK$^+$VGluT3$^+$INTS exhibit unique participation in, and regulation of, hippocampal network oscillations and place cell spatial information coding (*Del Pino et al., 2017*; *Lasztóczi et al., 2011*). However, direct functional evidence supporting glutamate/GABA corelease from CCK$^+$VGluT3$^+$INTS is currently lacking. In fact, focused cellular/network interrogation of CCK$^+$-VGluT3$^+$INTS has lagged other interneuron subtypes due to the notorious heterogeneity of CCK$^+$ interneurons making it difficult to reliably target CCK$^+$VGluT3$^+$INTS (*Fuzik et al., 2016*; *Harris et al., 2018*; *Klausberger and Somogyi, 2008*; *Kohus et al., 2016*; *Pelkey et al., 2017*; *Szabó et al., 2014*; *Tricoire et al., 2011*). Here we employed a VGluT3-Cre transgenic mouse driver line in combination with various conditional reporter lines and viral constructs to selectively investigate CCK$^+$VGluT3$^+$INTS with emphasis on their synaptic output properties. Using complementary paired recording and optogenetic approaches, we directly demonstrate CCK$^+$VGluT3$^+$INT-mediated GABA/glutamate cotransmission onto postsynaptic principal cell targets through GABA$_A$ and AMPA receptors, respectively. Under normal conditions, GABAergic inhibition dominates CCK$^+$VGluT3$^+$INT signaling to suppress circuit activity. However, glutamatergic transmission predominates following acute inhibition of GAD or dietary restriction of Vitamin B6, an essential cofactor for GAD enzymatic function. The enhanced glutamatergic tone from CCK$^+$VGluT3$^+$INTS under conditions of

compromised GAD function may exacerbate excitation/inhibition imbalances associated with various neuropsychatric disorders, including vitamin B6 deficiency which is known to enhance seizure susceptibility in rodents and humans (*Sharma et al., 1994*; *Tews, 1969*).

## Results and discussion

### Delayed onset of VGluT3 expression within CCK[+] interneurons

Closely related to the occurrence of corelease is the phenomenon of neurotransmitter respecification or switching involving the acquisition of one neurotransmitter and loss of another, often in developmentally regulated, graded, and transient fashions (*Spitzer, 2017*). For instance, VGluT3 is transiently expressed by inhibitory neurons in the developing auditory system and the glutamatergic phenotype is critical for proper tonotopic mapping for interaural sound localization (*Gillespie et al., 2005*; *Noh et al., 2010*). In contrast, native VGluT3 expression within hippocampal interneuron terminals is minimal during early development (postnatal day 5, P5), but progressively increases during juvenile to adult stages (*Figure 1A*, see also *Gras et al., 2005*). Importantly, we limited our initial VGluT3 immunolocalization analyses to terminals innervating the perisomatic regions of CA1 and CA3 pyramidal cells (PCs), the innervation domain of GABAergic BCs, to minimize any confounding influence of VGluT3 expressing extrahippocampal projection systems such as serotonergic median raphe inputs (*Gras et al., 2002*; *Somogyi et al., 2004*; *Varga et al., 2009*). Indeed, high-resolution imaging within CA1 stratum pyramidale (s.p.) confirmed VGluT3 immunosignal within presynaptic boutons delineated by strong cannabinoid receptor 1 (CBR1) labeling, a signature of CCK[+]BC terminals (*Figure 1B*). In addition, s.p. VGluT3 signal exhibited essentially complete overlap with VGAT and GAD67 (*Figure 1B*), confirming that the signal derives from local circuit interneurons. As a negative control, we confirmed that VGluT3 did not colocalize with synaptotagmin 2 (Syt2) containing terminals in s.p. which delineate perisomatic inputs from parvalbumin expressing BCs (*Figure 1B*).

In sum, these data illustrate a developmentally delayed onset for VGluT3 expression by CCK[+]-VGluT3[+]INTS and highlight the potential for GABA/glutamate corelease from the same presynaptic terminals as opposed to segregated GABA and glutamate release sites along individual axons (see for eg. *Fortin et al., 2019*; *Voisin et al., 2016*; *Zhang et al., 2015*). Moreover, evaluation of multiple boutons along individual parent axons suggests uniform distribution of VGluT3 at all terminals of VGluT3[+] interneurons minimizing the potential for postsynaptic target cell specificity to any glutamatergic phenotype of CCK[+]VGluT3[+]INTS (*Figure 1C–D*). While our data do not address the potential for segregated vesicle pools within individual CCK[+]VGluT3[+]INTS terminals (*Root et al., 2018*), prior studies have provided biochemical and ultrastructural evidence supporting VGluT3 and VGAT coexpression, as well as glutamate accumulation, within individual SVs from perisomatic targeting terminals (*Fasano et al., 2017*; *Stensrud et al., 2013*; *Stensrud et al., 2015*).

Developmental characterization of hippocampal CA1/CA3 cells reported by red fluorescent protein expression (RFP[+]) in VGluT3-Cre:Ai14 transgenic mice recapitulated the delayed onset of native VGluT3 expression with low cell densities reported early postnatally that increase and plateau through juvenile stages into adulthood (*Figure 2A*). Throughout postnatal development, but particularly at the earliest stages, Cajal Retzius cells within the dentate molecular layer along the hippocampal fissure are also reported (*Figure 2A* and *Figure 2—figure supplement 1A*). In addition, within the neocortex a subpopulation of deep layer RFP[+] pyramidal cells is observed, consistent with recently described '*Slc17a8*[+] L5 pyramidal cells' (*Figure 2—figure supplement 1B*; *Tasic et al., 2018*). However, within the mature hippocampus a sparse population of cells with somata distributed across strata oriens, pyramidale, and radiatum (s.o., s.p., s.r. respectively) of CA1/3 accumulates, similar to the heterogeneous distribution of hippocampal CCK[+]BCs (*Pelkey et al., 2017*). Indeed, within CA1 we observed excellent registration between the distributions of VGluT3-Cre: Ai14 reported and CCK immunoreactive cells and confirmed that the vast majority of genetically identified cells are CCK[+] (~80%; *Figure 2B*). Conversely, VGluT3-Cre:Ai14 reported cells comprised almost half of all CCK[+] CA1 interneurons, considerably higher than original estimates in rat (~25%) but similar to estimates in mouse (~46%) based on immunohistochemical (IHC) studies (*Del Pino et al., 2017*; *Somogyi et al., 2004*).

The specificity of the VGluT3-Cre:Ai14 line was further evidenced by the lack of overlap between RFP[+] cells with vasoactive intestinal peptide (VIP) signal (*Figure 2B*), consistent with the mutual

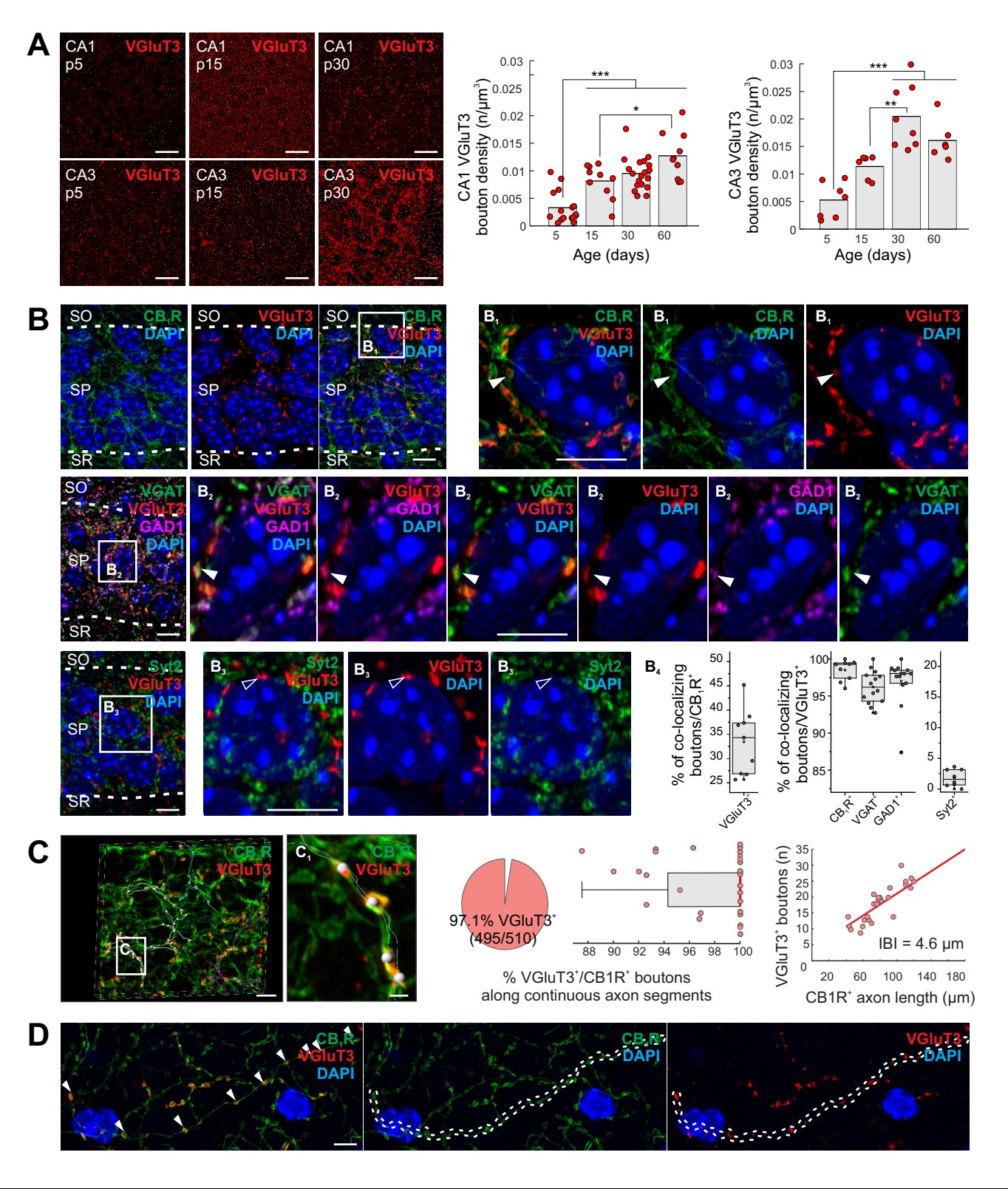

**Figure 1.** Delayed onset of VGluT3 expression in hippocampal CCK[+]VGluT3[+]INTS. (**A**) Representative images illustrating native VGluT3 expression within CA1/3 stratum pyramidale (SP) at postnatal day 5, 15, and 30 (p5,15,30) as indicated. Group data for terminal density counts in CA1 and CA3 throughout development are plotted at right (n = 6–19 sections from three to six animals per age were examined; individual observations (n) along with

*Figure 1 continued on next page*

*Figure 1 continued*

means are provided). (B) Representative high resolution Airyscan images illustrating colocalization of VGluT3 signal in CA1 perisomatic axon terminals with CB1R signal (upper row, ($B_1$)) as well as GAD1 and VGAT signals (second row, ($B_2$)) but not Syt2 labelled terminals (third row, ($B_3$)); SO, stratum oriens; SR, stratum radiatum). Widefield as well as digitally magnified images from boxed regions are provided showing merged and split channels for each set of markers, with DAPI labeling, as indicated. Filled arrowheads highlight terminals showing colocalization while open arrowheads highlight terminals lacking colocalization. Colocalization of VGluT3 with each marker is quantified in the box plots ($B_4$) with data normalized to total CB1R$^+$ terminal counts (left plot) or total VGluT3$^+$ terminal counts (middle and right plots; 1200–6580 terminals from n = 8–14 sections in each staining combination, from three mice were evaluated). (C) Representative Airyscan images and group data illustrating conserved VGluT3 expression at all putative release sites along continuous individual axon segments. Three individual axon segments are highlighted in the CA1 SP widefield image and a segment from one of these axons (boxed region, ($C_1$)) is digitally magnified. Group data plotted at right summarize the percentage of putative release sites along individual axons from CB1R$^+$/VGluT3$^+$ cells that are VGluT3$^+$ (pie chart and box plot, normalized to number of boutons in each axon segment) as well as the correlation between total number of boutons scored per length of each continuous axon examined (IBI, interbouton interval; 510 boutons were evaluated from n = 28 axon segments across three mice). (D) An additional example of conserved VGluT3 expression across all putative release sites of a continuous axon segment within SO with merged and split channels as indicated. For statistical analysis *p<0.05, **p<0.01, ***p<0.001. Scalebar in A = 10 µm; in B, C, D = 5 µm; in $C_1$ = 1 µm.

The online version of this article includes the following source data for figure 1:

**Source data 1.** Data plotted in *Figure 1*.

exclusion of VGluT3 and VIP by distinct subsets of CCK$^+$BCs (*Somogyi et al., 2004*). Similarly, RFP$^+$ cells rarely exhibited PV immunoreactivity (*Figure 2B*), consistent with the lack of VGluT3/Syt2 colocalization in perisomatic targeting terminals (*Figure 1B*). Unexpectedly, we regularly observed somatostatin (Som) expression within a minority (~20%) of RFP$^+$ cells, particularly within s.o. (*Figure 2B*). As VGluT3 expression has never been reported in Som interneurons this raised the concern of potential ectopic expression within this transgenic allele. However, we validated native VGluT3 mRNA expression within a small subset of Som interneurons of the mature hippocampus and confirmed active Cre in Som$^+$ cells reported in adult VGluT3-Cre:Ai14 mice (*Figure 2—figure supplement 1C*; see also *Sst.Erbb4.RGS10* and *Sst.Pnoc.Pvalb* single-cell RNASeq-based clusters in *Harris et al. (2018)*).

Taken together, our IHC findings confirm the relative specificity of the VGluT3-Cre driver line for identifying and genetically accessing hippocampal CCK$^+$VGluT3$^+$INTS and additionally reveal novel minority populations of VGluT3 expressing/reported neurons in the mature hippocampus including Cajal Retzius cells and a subset of Som interneurons. The labeling of postnatal Cajal Retzius cells likely reflects a transient period of VGluT3 expression by progenitors as the glutamatergic phenotype of mature Cajal Retzius cells relies exclusively on VGluT2 expression (*Tasic et al., 2018*) while VGluT3 expression is detected in subventricular zone progenitors (*Sánchez-Mendoza et al., 2017*). Importantly, the temporal profile of interneuron reporting faithfully recapitulates normal delayed VGluT3 production and indicates that functional interrogation of physiological roles for VGluT3 within these cells is best performed in mature animals beyond 1 month of age.

Interneuron migration into the rodent hippocampus is essentially complete by birth following which their densities decline through a combination of programmed cell death and volume expansion ultimately stabilizing between the second and third month postnatal (*Tricoire et al., 2011*). Thus, the late onset of VGluT3 expression in a subset of hippocampal interneurons likely reflects upregulation during maturation within hippocampal resident cells rather than late stage infiltration and sudden axon extension by VGluT3 expressing interneurons. Indeed, a high proportion of VGluT3 lacking RFP$^+$ perisomatic terminals in young VGluT3-Cre:Ai14 mice is consistent with delayed VGluT3 accumulation in preexisting axons (*Figure 2—figure supplement 2A*). Moreover, the proportion of VGluT3$^+$ CB1R$^+$ perisomatic terminals significantly increases from postnatal days 10 to 90 with no significant increase in overall CB1R$^+$ terminal density consistent with delayed accumulation of VGluT3 in a subset of hippocampal resident CCK$^+$BCs (*Figure 2—figure supplement 2B*).

## CCK$^+$VGluT3$^+$INTS exhibit rich anatomical diversity and typical GABAergic transmission properties

VGluT3 expression within cortical interneurons was originally described as limited to a subset of CCK$^+$BCs (*Somogyi et al., 2004*) and VGluT3 protein/mRNA content is typically interpreted within this context (e.g. *Bezaire and Soltesz, 2013*; *Del Pino et al., 2017*; *Harris et al., 2018*). However,

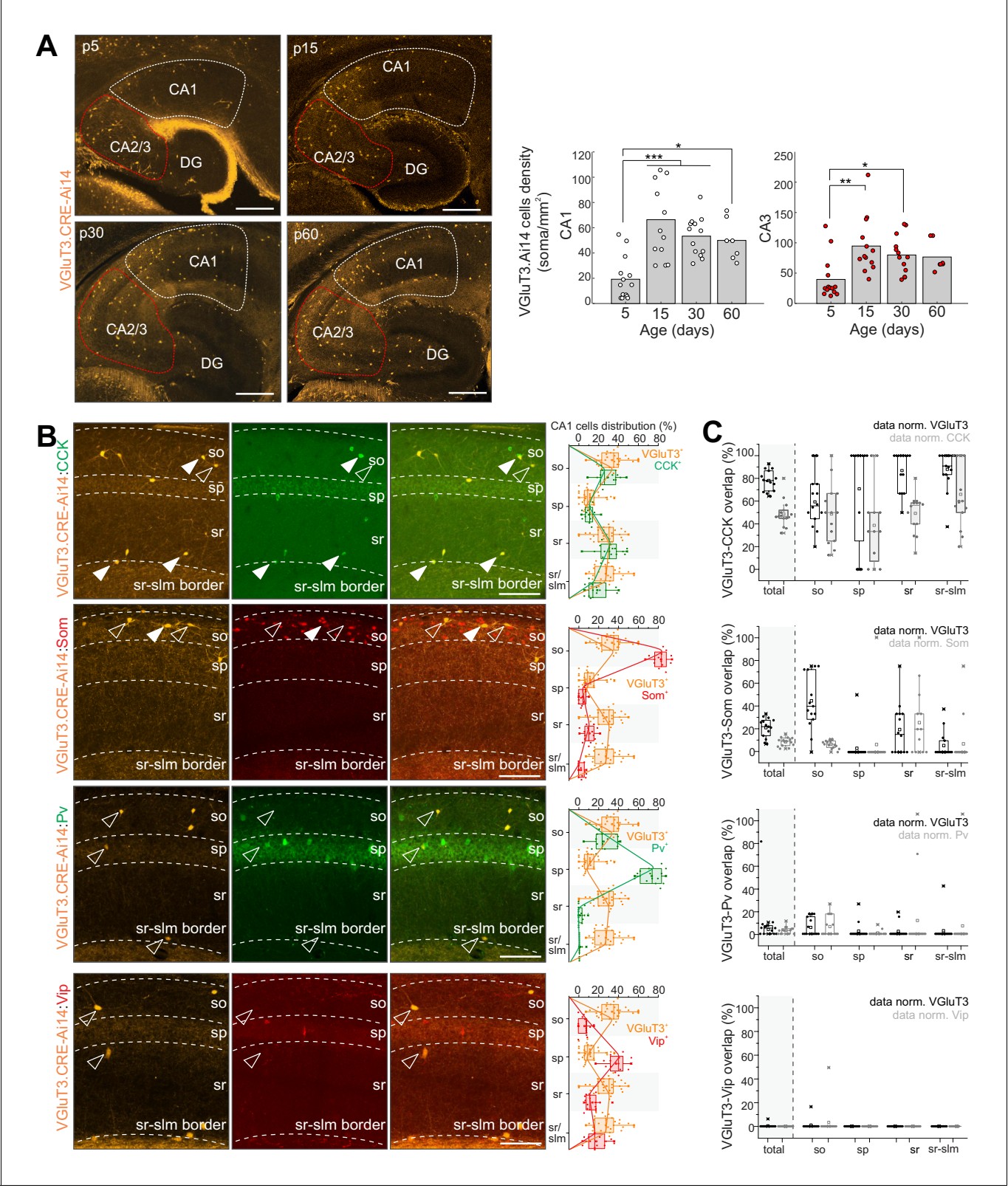

**Figure 2.** Developmental and IHC evaluation of hippocampal VGluT3-Cre:Ai14 reported cells. (**A**) Representative images (left) and group data summary (right) illustrating the developmental profile of RFP[+] cell densities in CA1 and CA2/CA3 regions of the hippocampus as indicated (n = 7–14 sections from three to six mice for each time point were evaluated; DG, dentate gyrus). (**B**) Representative images of CA1 used to evaluate colocalization of RFP[+] cells with CCK, Som, PV, and VIP as indicated (left). Merged and split channels are shown for each marker pair with filled arrowheads highlighting

*Figure 2 continued on next page*

Figure 2 continued

sample cells displaying colocalized signal and open arrowheads highlighting sample cells without colocalization. Distributions of labelled cells for each interneuron marker within CA1 from s.o. to s.l.m. (stratum lacunosum moleculare) are plotted for comparison to RFP$^+$ cell distributions (right). (C) The degree of overlap between cells expressing each interneuron marker with those expressing RFP is provided for all of CA1 as well as for each sublayer. Data are plotted normalized to either the RFP$^+$ cell population (black) or the given interneuron marker population (gray; 336–619 cells from n = 9–15 sections from three mice were evaluated per staining combination). Scalebar in A = 500 μm; in B = 100 μm.

The online version of this article includes the following source data and figure supplement(s) for figure 2:

**Source data 1.** Data plotted in *Figure 2*.
**Figure supplement 1.** Further characterization of RFP$^+$ cell populations reported in VGluT3-Cre:Ai14 mice.
**Figure supplement 2.** Further developmental characterization of VGluT3$^+$ terminals and VGluT3$^+$ BC anatomies.
**Figure supplement 2—source data 1.** Data plotted in *Figure 2—figure supplement 2*.

evaluation of individual RFP$^+$ interneurons from VGluT3-Cre:Ai14 mice revealed a much broader diversity of cell morphologies. Post-hoc recovered anatomies following electrophysiological recordings from CA1/3 RFP$^+$ cells in acute slices revealed BCs as well as diverse dendrite targeting interneurons (DTIs) such as Schaffer collateral associated cells, perforant path associated cells, and mossy fiber associated cells (*Figure 3A*; see also *Lasztóczi et al., 2011*). Interestingly, VGluT3-Cre:Ai14 reported BCs displayed mature axon profiles as early as P13 supporting the conclusion of delayed VGluT3 expression within anatomically mature BCs rather than delayed axon extension following VGluT3 expression (*Figure 2—figure supplement 2C*). Consistent with VGluT3 expressing DTIs we observed that a subset of VGluT3$^+$ terminals within CA1 s.r. colocalized with CB1R, and VGAT (*Figure 3—figure supplement 1*). In contrast, a population of large VGluT3$^+$ terminals, particularly enriched along the s.r.-stratum lacunosum moleculare (s.l.m.) border, were CB1R$^-$/VGAT$^-$ but often expressed vesicular monoamine transporter 2 (VMAT2$^+$) consistent with raphe serotonergic fibers (*Figure 3—figure supplement 1*). Thus, the VGluT3$^+$ subset of CA1/3 CCK$^+$ interneurons extends beyond BCs encompassing the entire known spectrum of CCK$^+$ interneurons (reviewed in *Pelkey et al., 2017*). This underappreciated diversity is reflected within transcriptional profiling as VGluT3 signature is prominent within at least six distinct clusters of CCK$^+$ interneurons (*Harris et al., 2018*).

Despite their considerable anatomical heterogeneity, CCK$^+$VGluT3$^+$INTS exhibited remarkably homogeneous active and passive membrane properties. Both BCs and DTIs consistently displayed high input resistances, slow time constants, hyperpolarization-induced sag, and regular accommodating firing patterns typical of CCK$^+$ interneurons (e.g. traces in *Figure 3A*). Indeed, we failed to detect any significant differences between VGluT3$^+$ BCs and DTIs across a broad range of biophysical features (*Table 1*). Thus, CCK$^+$VGluT3$^+$ BCs and DTIs appear indistinguishable with respect to basic intrinsic electrophysiological properties (see also *Kohus et al., 2016*; *Oláh et al., 2019*; *Rovira-Esteban et al., 2017*; *Szabó et al., 2014*).

Next, we examined unitary inhibitory postsynaptic currents (uIPSCs) in paired recordings between synaptically coupled CCK$^+$VGluT3$^+$INTS and CA1PCs. CCK$^+$VGluT3$^+$INT-CA1PC pairs yielded uIPSCs with high inter-event amplitude variability, modest short-term depression, and train-induced asynchronous release (*Figure 3B–C*). Moreover, transmission from CCK$^+$VGluT3$^+$INTS was susceptible to depolarization-induced suppression of inhibition (DSI), consistent with the expression of CB1Rs within presynaptic terminals of VGluT3$^+$ BCs and DTIs (*Figure 3D–E*). Thus, CCK$^+$VGluT3$^+$-INTS reported in VGluT3-Cre:Ai14 mice exhibit all the hallmark features expected for CCK$^+$ interneuron-mediated inhibition (*Armstrong and Soltesz, 2012*; *Daw et al., 2009*; *Glickfeld and Scanziani, 2006*; *Hefft and Jonas, 2005*; *Wyeth et al., 2017*). Interestingly, CCK$^+$VGluT3$^+$INT-CA1PC pairs yielded amplitudes/potencies within the low range of those reported in prior studies of CCK$^+$ interneurons that did not selectively target the VGluT3$^+$ subsets (*Daw et al., 2009*; *Kohus et al., 2016*; *Lee et al., 2010*; *Vargish et al., 2017*; *Wyeth et al., 2017*); note that in some cases differences in Cl$^-$ driving force must be accounted for). Thus, CCK$^+$VGluT3$^+$INTS do not appear to exhibit enhanced GABA release compared to the general population of CCK$^+$ BCs and DTIs, arguing against a role for VGluT3 in facilitating SV GABA filling through 'vesicular synergy' (*Amilhon et al., 2010*; *El Mestikawy et al., 2011*; *Fasano et al., 2017*; *Frahm et al., 2015*; *Gras et al., 2008*; *Hnasko et al., 2010*). Similarly, a recent study in the amygdala failed to observe enhanced unitary transmission from VGluT3$^+$ CCK interneurons over those lacking VGluT3 (*Rovira-Esteban et al.,*

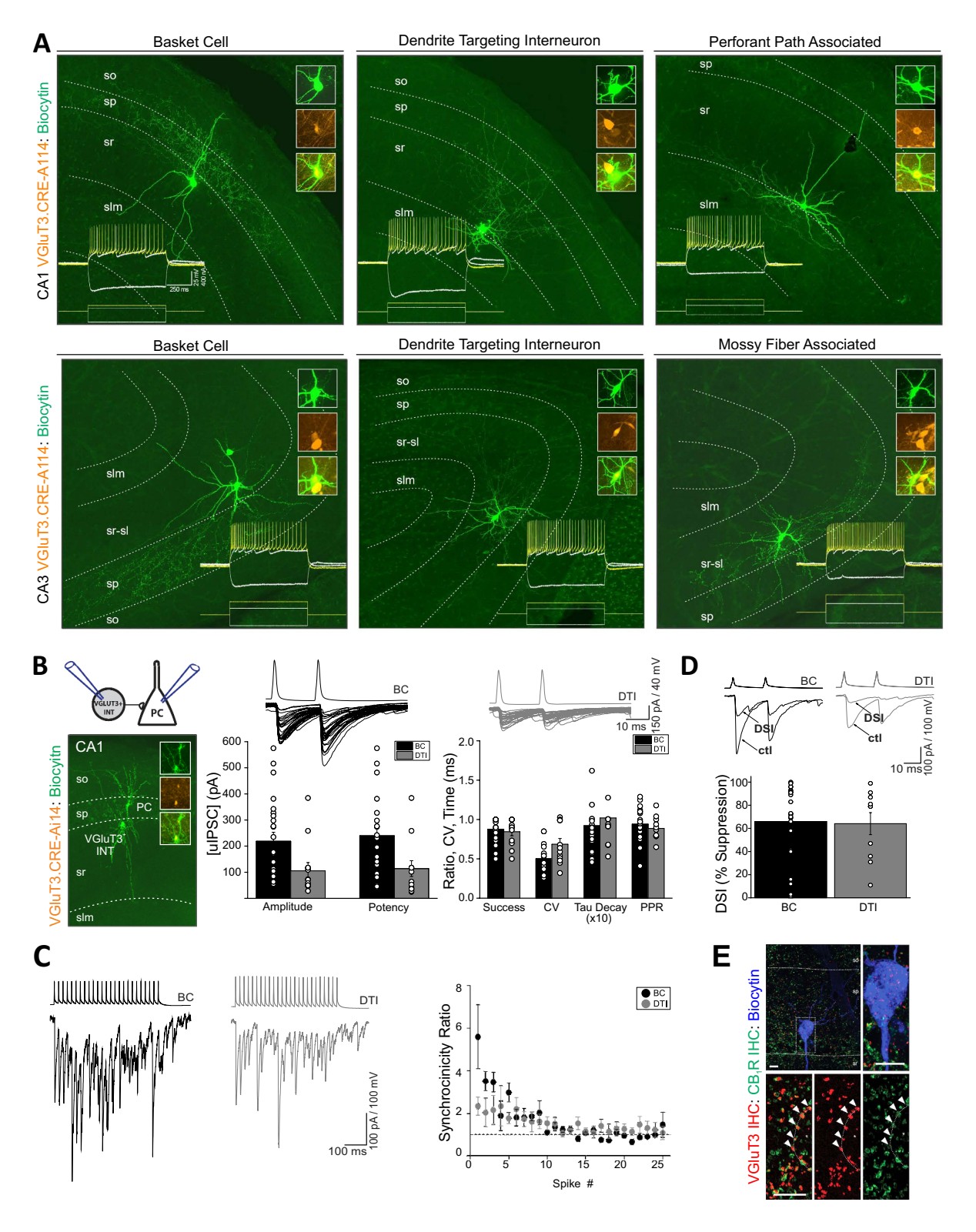

**Figure 3.** Anatomical, electrophysiological, and basic synaptic properties of hippocampal CCK+VGluT3+INTs. (A) Images of representative RFP+ cells recorded in VGluT3-Cre:Ai14 mice from CA1 (upper row) and CA3 (lower row) hippocampus illustrating prototypical BCs and DTIs. Inset traces illustrate membrane responses of each representative cell (upper series of traces) to the current injection waveforms shown (lower series of traces). Image insets highlight colocalization of the somatic dye fill (biocytin) with RFP. (B) Basic properties of unitary GABAergic synaptic transmission between VGluT3+BCs

*Figure 3 continued on next page*

*Figure 3 continued*

(black) or VGluT3⁺DTIs (gray) with CA1 PCs. At left, the recording configuration is schematized above an image of a sample RFP⁺BC-CA1PC pair. At right, sample traces show individual trial synaptic events in postsynaptic CA1 PCs (lower traces) following presynaptic spiking (upper traces) in either RFP⁺ BCs or DTIs as indicated. The basic properties of transmission are summarized in the group data bar charts below (n = 25 and 12 pairs, from 17 and 10 mice for BC and DTI pairs, respectively). Note that for these recordings postsynaptic CA1PC cells were filled with high [Cl⁻] and held at −70 mV yielding GABAergic dominated inward currents. (C) Sample traces (left) from representative recordings of VGluT3⁺BC-CA1PC and VGluT3⁺DTI-CA1PC pairs as indicated illustrating asynchronous release of GABA during presynaptic trains of stimuli. Note the postsynaptic events are tightly synchronized with presynaptic APs early in the train and then fail to remain time locked to presynaptic APs as the train continues resulting in a progressive reduction in the synchronicity ratio plotted at right for all recordings (n = 6 and 5 pairs from 4 and 4 mice for BC and DTI pairs, respectively). (D) Sample traces (above) and group data bar chart summary (below) illustrating DSI of VGluT3⁺BC/DTI-CA1PC pairs. For each set of traces an averaged uIPSC obtained under control conditions (ctl) is shown in comparison with an averaged uIPSC obtained following depolarization of postsynaptic CA1PCs to 0 mV for 5 s (DSI, 5–10 individual uIPSCs were averaged for each condition). The group data bar chart quantifies the magnitude of uIPSC suppression induced by DSI (n = 22 and 10 pairs from 17 and 9 mice for BC and DTI pairs respectively). (E) Representative sample image of a biocytin filled postsynaptic CA1PC soma decorated with VGluT3⁺CB1R⁺ presynaptic terminals consistent with DSI of VGluT3⁺BC /-CA1 PC pairs. Throughout the figure summary plots provide individual observations as well as group means ± SEM.

The online version of this article includes the following source data and figure supplement(s) for figure 3:

**Source data 1.** Data plotted in *Figure 3*.
**Figure supplement 1.** Evaluation of dendrite targeting VGluT3⁺ terminals.

*2017*) and forced VGluT3 expression in cultured striatal GABAergic interneurons failed to facilitate quantal GABA transmission (*Zimmermann et al., 2015*). In addition to vesicular synergy, VGluT3 expression has been hypothesized to allow glutamate release from CCK⁺VGluT3⁺INTS to enhance DSI through activation of postsynaptic metabotropic glutamate receptor subtype 5 (mGluR5) activation for facilitated endocannabinoid production (*Omiya et al., 2015*). However, DSI sensitivity of release from CCK⁺VGluT3⁺INTS is comparable to that of the general CCK⁺ interneuron population (*Daw et al., 2009*; *Glickfeld and Scanziani, 2006*; *Vargish et al., 2017*; *Wyeth et al., 2017*).

**Table 1.** CCK⁺VGluT3⁺INTs membrane properties.

| Active/Passive Membrane Properties | VGluT3⁺ BCs (mean ± SEM, n = 30–37 observations/ parameter, from 37 cells) | VGluT3⁺ DTIs (mean ± SEM, n = 24–26 observations/ parameter, from 26 cells) |
|---|---|---|
| Resting potential (mV) | −56.9 ± 0.8 | −59.0 ± 1.1 |
| Input resistance (MΩ) | 158.5 ± 9.9 | 149.7 ± 10.1 |
| Time constant (ms) | 20.8 ± 1.1 | 22.0 ± 1.0 |
| Capacitance (pF) | 135.7 ± 5.8 | 150.1 ± 6.3 |
| Sag index | 0.83 ± 0.01 | 0.82 ± 0.02 |
| Spike threshold (mV) | −41.9 ± 0.8 | −39.86 ± 1.1 |
| Spike amplitude (mV) | 57.4 ± 1.7 | 55.7 ± 1.9 |
| Spike half-width (ms) | 0.74 ± 0.03 | 0.80 ± 0.05 |
| Maximal rise slope (mV/ms) | 197.7 ± 8.5 | 186.4 ± 13.9 |
| Maximal decay slope (mV/ms) | −87.8 ± 4.4 | −81.1 ± 7.2 |
| Frequency at 2x Threshold (Hz) | 34.9 ± 2.4 | 31.3 ± 1.8 |
| Adaptation ratio at 2x threshold | 0.43 ± 0.04 | 0.43 ± 0.05 |
| AHP amplitude (mV) | −10.3 ± 0.6 | −11.8 ± 0.9 |

The online version of this article includes the following source data for Table 1:
**Source data 1.** Individual neuron properties.

# Glutamate release from CCK⁺VGluT3⁺INTS activates postsynaptic AMPARs

Although the perisomatic domain of hippocampal PCs is devoid of conventional excitatory synapses (*Megías et al., 2001*), functional α-amino-3-hydroxy-5-methyl-4-isoxazolepropionic acid receptors (AMPARs) are abundantly expressed within PC somatic membranes (*Figure 4—figure supplement 1A*; and see for eg. *Andrasfalvy and Magee, 2001*; *Jonas and Sakmann, 1992*; *Lu et al., 2009*). Thus, we reasoned that any potential glutamate release from CCK⁺VGluT3⁺INTS, particularly the BC subset, should be electrophysiologically detectable through AMPAR activation. Indeed, forced expression of VGluT1/2 or 3 within cultured GABAergic interneurons leads to glutamate corelease that reliably activates AMPARs at autaptic inputs with kinetics approaching conventional excitatory synapses (*Takamori et al., 2000*; *Takamori et al., 2001*; *Zimmermann et al., 2015*).

Our initial paired recording data indicated that GABAergic transmission dominates CCK⁺-VGluT3⁺INT output under normal conditions (*Figure 3B*, note the slow kinetics of postsynaptic events). Thus, to increase our chances of observing an AMPAR-mediated component to transmission, we first employed an optogenetic strategy to simultaneously activate many CCK⁺VGluT3⁺INT terminals. Channelrhodopsin2 (ChR2), or the Chronos variant, were expressed within CCK⁺VGluT3⁺-INTS by either crossing VGluT3-Cre mice with a Cre-dependent ChR2 line (VGluT3-Cre:Ai32) or by stereotactic delivery of AAV:Flex-Chronos-GFP into the hippocampus of VGluT3-Cre mice (VGluT3-Cre:Chronos), respectively (*Figure 4A*). We made whole-cell recordings from CA1 PCs setting the chloride reversal potential at ∼−60 mV while voltage clamping the membrane potential depolarized to this level (at −30 to −50 mV). In this configuration, GABAergic transmission yields outward currents while AMPAR-mediated transmission generates inward currents allowing for unambiguous dissociation of GABAergic and glutamatergic signaling based on the polarity of synaptic events. Both optogenetic strategies yielded large light-evoked outward synaptic currents under control conditions at the beginning of all recordings confirming GABA as the dominant transmitter (*Figure 4B–C*). However, pharmacological inhibition of GABARs with a combination of picrotoxin, bicuculline, and CGP 55845 (pic/bic/CGP) consistently uncovered a residual small inward synaptic conductance (*Figure 4B–C*). Consistent with mediation by AMPARs, the residual light-evoked inward synaptic currents exhibited linear current-voltage relations with reversal near 0 mV and rapid decay kinetics compared to the corresponding GABAergic events (*Figure 4D–E*). Moreover, the pic/bic/CGP-insensitive light-evoked events displayed pharmacological profiles typical of AMPARs including prolonged decays following block of receptor desensitization with cyclothiazide (CTZ) and antagonism by DNQX, GYKI, or γ-DGG (*Figure 4E–F*).

To confirm that the glutamatergic phenotype observed in VGluT3-Cre:Ai32 and VGluT3-Cre:Chronos mice derives specifically from CCK⁺VGluT3⁺INTS, we performed several additional analyses and experiments. Importantly, no glutamatergic component to transmission was associated with light-evoked transmission in Nkx2.1-Cre:Ai32 mice that express ChR2 in medial ganglionic eminence derived interneurons which excludes CCK⁺ cells, confirming that light activation of hippocampal interneurons does not generally promote glutamate release nonspecifically (*Figure 4—figure supplement 1B*). In VGluT3-Cre:Ai32 mice light-sensitive terminals from any VGluT3⁺ projection system entering the hippocampus could potentially provide a confounding source for the observed glutamatergic phenotype (e.g. raphe projections known to release glutamate, *Varga et al., 2009*). However, the perfect register between GABA and glutamate PSC latencies is consistent with a common presynaptic source for both transmitters (*Figure 4B* inset). Moreover, the complete overlap between VGluT3-Cre:Ai32 and VGluT3-Cre:Chronos data sets argues in favor of local hippocampal CCK⁺-VGluT3⁺INTS in mediating both responses (*Figure 4B–I*). Indeed, the GABAergic and glutamatergic light-evoked synaptic events in VGluT3Cre:Ai32 and VGluT3Cre:Chronos mice exhibit prominent DSI as expected for synaptic events derived from CCK⁺VGluT3⁺INTS endowed with presynaptic CB1Rs (*Figure 4G–I*). Although Cajal Retzius cells provide a potential confounding local glutamatergic source in our optogenetic strategy (*Figure 2A*) these cells rarely provide direct synaptic input to CA1 PCs (*Quattrocolo and Maccaferri, 2014*). Moreover, the short latencies and rapid kinetics of our AMPAR-mediated events are inconsistent with distal excitation on apical dendrite tufts within s.l. m.

In theory, the GABA and glutamate release observed in our optogenetic experiments could derive from distinct CCK⁺VGluT3⁺INT cohorts with overlapping innervation patterns coincidentally

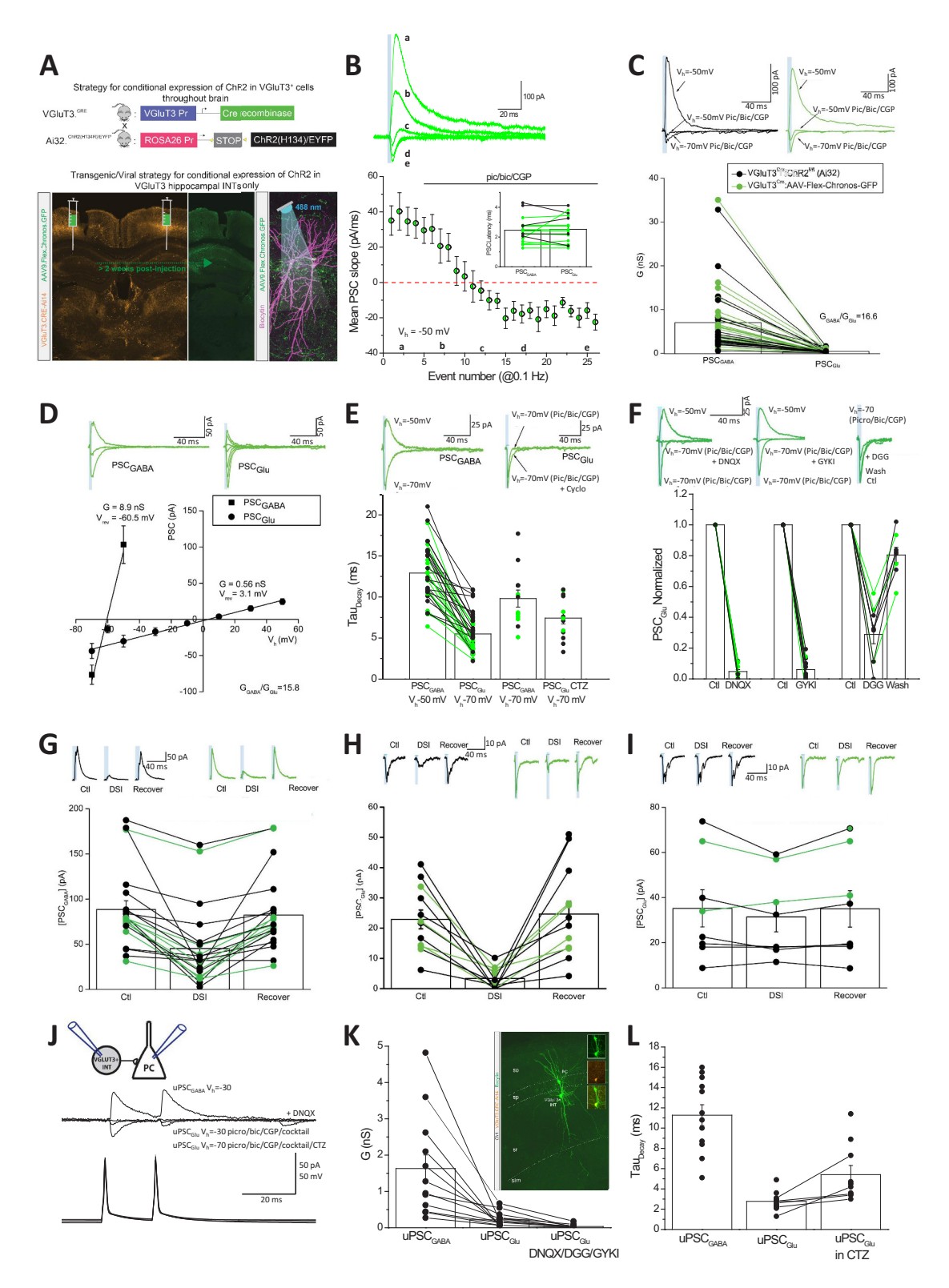

**Figure 4.** Glutamate and GABA corelease from CCK[+]VGluT3[+]INTs. (**A**) Schematics depicting two strategies to genetically introduce ChR2 or the Chronos variant within CCK[+]VGluT3[+]INTs by either breeding VGluT3-Cre:Ai32 mice (above) or introducing AAV-CAG-Flex-Chronos-GFP selectively into the hippocampus of VGluT3-Cre mice (below) respectively. In both cases, the optogenetic actuator is driven by illumination with blue light while recording CA1PCs as illustrated in the bottom right subpanel. Here and throughout data from both strategies have been pooled and individual

*Figure 4 continued on next page*

*Figure 4 continued*

recordings from the VGluT3-Cre:Ai32 and VGluT3:Chronos strategies are depicted in black and green respectively. (B) Traces from a representative recording (above) and group data summary (below) for a subset of recordings illustrating the conversion of an outward GABAergic dominated postsynaptic current (PSC$_{GABA}$) to an inward glutamatergic mediated synaptic event (PSC$_{Glu}$) following introduction of GABA$_{A/B}$ receptor antagonists as indicated (n = 16 recordings from nine mice). Traces are average events (3–5 individual sweeps) obtained at the times indicated in the group plot below. Note that Cl$^-$ reversal potential is set to ~60 mV and postsynaptic cells are initially held positive to this so that GABAergic events are outward (positive initial slope) while glutamatergic events are inward (negative initial slope). Bar chart inset summarizes the latencies of GABAergic and pharmacologically isolated glutamatergic events for each recording making up the time course plot. (C) Averaged traces from representative recordings (above) and group data summary (below) comparing GABAergic (PSC$_{GABA}$) and glutamatergic (PSC$_{Glu}$) synaptic conductances (G, here and throughout corresponding conductances measured for each individual recording are connected in the group data plot, n = 49 recordings from 22 mice). (D) Traces from representative recordings (above) and group data plots illustrating the current-voltage relations of GABAergic and glutamatergic synaptic events as indicated (PSC$_{GABA}$, n = 12 cells from four mice; PSC$_{Glu}$, n = 10 cells from four mice). (E) Traces from representative recordings (above) and group data (plotted below) highlighting the distinct kinetics of PSC$_{GABA}$ and PSC$_{Glu}$ (n = 35 cells from 15 mice with corresponding GABAergic and glutamatergic kinetics, connected points). Also shown are PSC$_{GABA}$ kinetics observed at −70 mV for inward GABAergic events (n = 12 cells from three mice) and PSC$_{Glu}$ kinetics obtained in CTZ (n = 13 cells from seven mice). (F) Traces from representative recordings (above) and group data summaries (below) illustrating antagonism of PSC$_{Glu}$ by DNQX (n = 12 cells from four mice), GYKI 53655 (n = 12 cells from seven mice) or γ-DGG (n = 9 cells from six mice, note reversible block following wash). Group data are expressed normalized to control responses obtained prior to antagonist application. (G–H) Traces from representative recordings (above) and group data summary plots (below) illustrating DSI sensitivity of PSC$_{GABA}$ (G, n = 21 cells from seven mice), and PSC$_{Glu}$ (H, n = 12 cells from seven mice). Absolute amplitudes of averaged PSCs before (Ctl) during (DSI) and after DSI (Recover) are plotted. (I) As for H but with BAPTA included in the recording pipette to block endocannabinoid production (n = 8 cells from three mice). (J) Averaged traces from a representative CCK$^+$VGluT3$^+$BC-PC pair recording (schematic inset) illustrating GABA and glutamate cotransmission. Antagonism of the outward unitary GABAergic event (uPSC$_{GABA}$) unmasks a smaller inward unitary glutamatergic event (uPSC$_{Glu}$) that is enhanced by CTZ and increased driving force and also sensitive to DNQX. (K) Group data summary of uPSC$_{GABA}$ and uPSC$_{Glu}$ conductances for all pairs exhibiting glutamatergic transmission (n = 12 and 17 pairs in 10 mice for uPSC$_{GABA}$ and uPSC$_{Glu}$ respectively; 12 recordings with corresponding uPSC$_{GABA}$ and uPSC$_{Glu}$ are connected by lines, in the remainder only uPSC$_{Glu}$ was evaluated as pair was obtained in partial or full GABA$_A$R antagonism). Also plotted is antagonism of uPSC$_{Glu}$ by DNQX or GYKI or DGG in a subset of recordings (n = 10 pairs in five mice). Inset image shows sample pair anatomical recovery. (L) Group data summary of the kinetic properties of uPSC$_{GABA}$ (n = 12 pairs) and uPSC$_{Glu}$ (n = 11) as well as uPSC$_{Glu}$ in CTZ (n = 10; in five cases with corresponding control and CTZ conditions as illustrated by the connected points). Throughout the figure summary plots provide individual observations as well as group means ± SEM.

The online version of this article includes the following source data and figure supplement(s) for figure 4:

**Source data 1.** Data plotted in *Figure 4*.
**Figure supplement 1.** Additional data in support of glutamatergic output from VGluT3$^+$ INTs.
**Figure supplement 1—source data 1.** Data plotted in *Figure 4—figure supplement 1*.

---

activated by the light pulses. Thus, to confirm GABA/glutamate cotransmission from individual CCK$^+$VGluT3$^+$INTS we revisited CCK$^+$VGluT3$^+$INTS-CA1PC paired recordings under conditions of reversed polarity for each transmitter species. To maximize our chances of detecting an AMPAR-mediated event, we coapplied antagonists for CB1Rs (AM251), kainate receptors (UBP 302), and metabotropic glutamate receptors (LY 341495) in combination with pic/bic/CGP to minimize the potential for tonic or use-dependent presynaptic depression of release (*Armstrong and Soltesz, 2012*; *Daw et al., 2010*; *Wyeth et al., 2017*). In addition, CTZ was frequently included to eliminate any potential for perisomatic AMPARs to accumulate in a desensitized state (*Barnes-Davies and Forsythe, 1995*; *Neher and Sakaba, 2001*). With this drug cocktail, optogenetically triggered glutamatergic transmission in VGluT3-Cre:Ai32 mice increased roughly three fold confirming the potential for enhancing transmission (*Figure 4—figure supplement 1C*). Using this strategy, we observed an AMPAR-mediated component in 17/53 synaptically coupled CCK$^+$VGluT3$^+$INT-CA1PC paired recordings (*Figure 4J–L*). In the majority of pairs exhibiting AMPAR-mediated transmission, we directly monitored the GABA to glutamate transition (12/17, connected points in *Figure 4K*; remainder obtained following pic/bic/CGP) unequivocally illustrating GABA and glutamate corelease from individual CCK$^+$VGluT3$^+$INTS onto the same postsynaptic partner.

## Compromising GAD function promotes the glutamatergic phenotype of CCK$^+$VGluT3$^+$INTS

Direct functional coupling between SV anchored GAD with VGAT promotes efficient GABA loading and is expected to minimize terminal glutamate substrate for VGluT3, consistent with our observations that CCK$^+$VGluT3$^+$INT transmission is dominated by GABA (*Jin et al., 2003*; *Somogyi, 2006*).

However, GAD and VGAT levels are plastic and may be homeostatically scaled according to network activity (*De Gois et al., 2005*; *Hanno-Iijima et al., 2015*). Moreover, GABA synthesis and SV loading can be compromised in various nervous system disorders including schizophrenia, depression, Huntington's Disease, and GAD autoantibody associated epilepsy (*de Jonge et al., 2017*; *Gupta et al., 2001*; *Hsu et al., 2018*; *Li et al., 2017*; *Ma et al., 2016*; *Mäkelä et al., 2018*; *Rush et al., 2012*; *Torrey et al., 2005*). To investigate the potential for amplification of glutamatergic output from CCK+VGluT3+INTS following disrupted GABA synthesis, we first treated slices from VGluT3-Cre: Ai32 and VGluT3:Chronos mice with semicarbazide (SCZ) to acutely inhibit GAD function (*Bell and Anderson, 1972*; *Killam and Bain, 1957*). Following 2–4 hr of SCZ incubation GABAergic output from CCK+VGluT3+INTS was severely compromised often allowing glutamatergic output to be observed even prior to pic/bic/CGP treatment (e.g. traces in *Figure 5A*). Overall, the impaired GABAergic signaling coupled with almost a doubling of the AMPAR-mediated synaptic conductance (evaluated in pic/bic/CGP plus the cocktail), yielded a dramatic reduction in the GABA/glutamate ratio of CCK+VGluT3+INT transmission (*Figure 5A*). Importantly, in SCZ-treated slices without AM 251 on board the glutamatergic phenotype of light-evoked events retained DSI sensitivity consistent with SCZ treatment promoting glutamate release from CB1R expressing CCK+VGluT3+INTS rather than uncovering a novel source (*Figure 5B*). Indeed, enhanced glutamatergic events in SCZ-treated slices could be inhibited by the CB1R agonist WIN-55212–2 at a high dose (5 µM) to overcome the antagonistic effects of AM 251 (2 µM) normally present in our cocktail (26.6 ± 0.1% of control, n = 5, p=0.04). Moreover, enhanced glutamatergic transmission was evident in a limited data set evaluated with CCK+VGluT3+INT-CA1PC paired recordings in SCZ treated slices (*Figure 5C*).

Vitamin B6, also known as pyridoxal phosphate (PLP), is an essential cofactor for GAD enzymatic function and B6 nutritional deficiency reduces GABA levels throughout the brain thereby enhancing seizure susceptibility in rodents and humans (*Sharma et al., 1994*; *Tews, 1969*). Indeed, B6 administration serves as an effective therapeutic strategy for neonatal epilepsy of unknown etiology (*Bessey et al., 1957*; *Grillo et al., 2001*; *Gupta et al., 2001*). To complement our acute in vitro GAD inhibition experiments with SCZ, we attempted to chronically compromise GABA synthesis in vivo by depleting PLP in VGluT3-Cre:Ai32 mice through a B6-deficient diet (B6⁻; *Sharma et al., 1994*). Optogenetic evaluation of mice kept on the B6⁻ diet for 3–9 months revealed reduced GABAergic transmission with increased glutamatergic transmission from CCK+VGluT3+INTS compared to littermates maintained on a control diet (B6⁺) leading to a significantly depressed GABA/glutamate synaptic conductance ratio (*Figure 5D*). Together our findings confirm that that the glutamatergic phenotype of CCK+VGluT3+INTS can be significantly enhanced under conditions of compromised GABA synthesis.

Given the privileged perisomatic input domain of many CCK+VGluT3+INTS close to the site of action potential initiation, glutamatergic signaling from these cells has unique potential to promote inappropriate network excitation. With inhibition intact (i.e. no GABA or GAD antagonists) optogenetic activation of slices from VGluT3Cre:Ai32 mice was highly effective in suppressing network activity (*Figure 5—figure supplement 1*). In contrast, in approximately 20% of the optogenetic recordings performed in our drug cocktail polysynaptic network barrages were observed following short duration single-pulse light-evoked monosynaptic signaling (e.g. *Figure 5E*). Such polysynaptic barrages are consistent with CCK+VGluT3+INT-driven suprathreshold activation of PCs (e.g. *Figure 5F*) that engage recurrent network activity. Importantly, polysynaptic network events observed in both VGluT3-Cre:Ai32 and VGluT3-Cre:Chronos mice were blocked by CB1R activation with WIN-55212–2 consistent with the events being triggered by CCK+VGluT3+INTS (*Figure 5E,G*). Moreover, all the basic features of optogenetically driven glutamatergic signaling observed in VGluT3-Cre:Ai32 and VGluT3-Cre:Chronos mice, including triggering of polysynaptic network activity, were observed in slices from VGluT3-Cre mice injected with a viral vector encoding Cre-dependent and interneuron-specific ChR2 (*Figure 5H–L*; VGluT3-Cre:AAV-mDlx-Flex-ChR2; *Dimidschstein et al., 2016*). Thus, CCK+VGluT3+INTS have the capacity to trigger paradoxical network excitation.

VGluT3 is considered atypical in that it is frequently expressed by neuronal populations that do not utilize glutamate as their primary transmitter. In many of these systems, VGluT3 imparts unique synaptic properties such as glutamate corelease and vesicular synergy (*Amilhon et al., 2010*; *Gras et al., 2008*; *Higley et al., 2011*; *Nelson et al., 2014*; *Noh et al., 2010*; *Sakae et al., 2015*; *Varga et al., 2009*; *Wang et al., 2019*). For almost 20 years, VGluT3 has been recognized as a

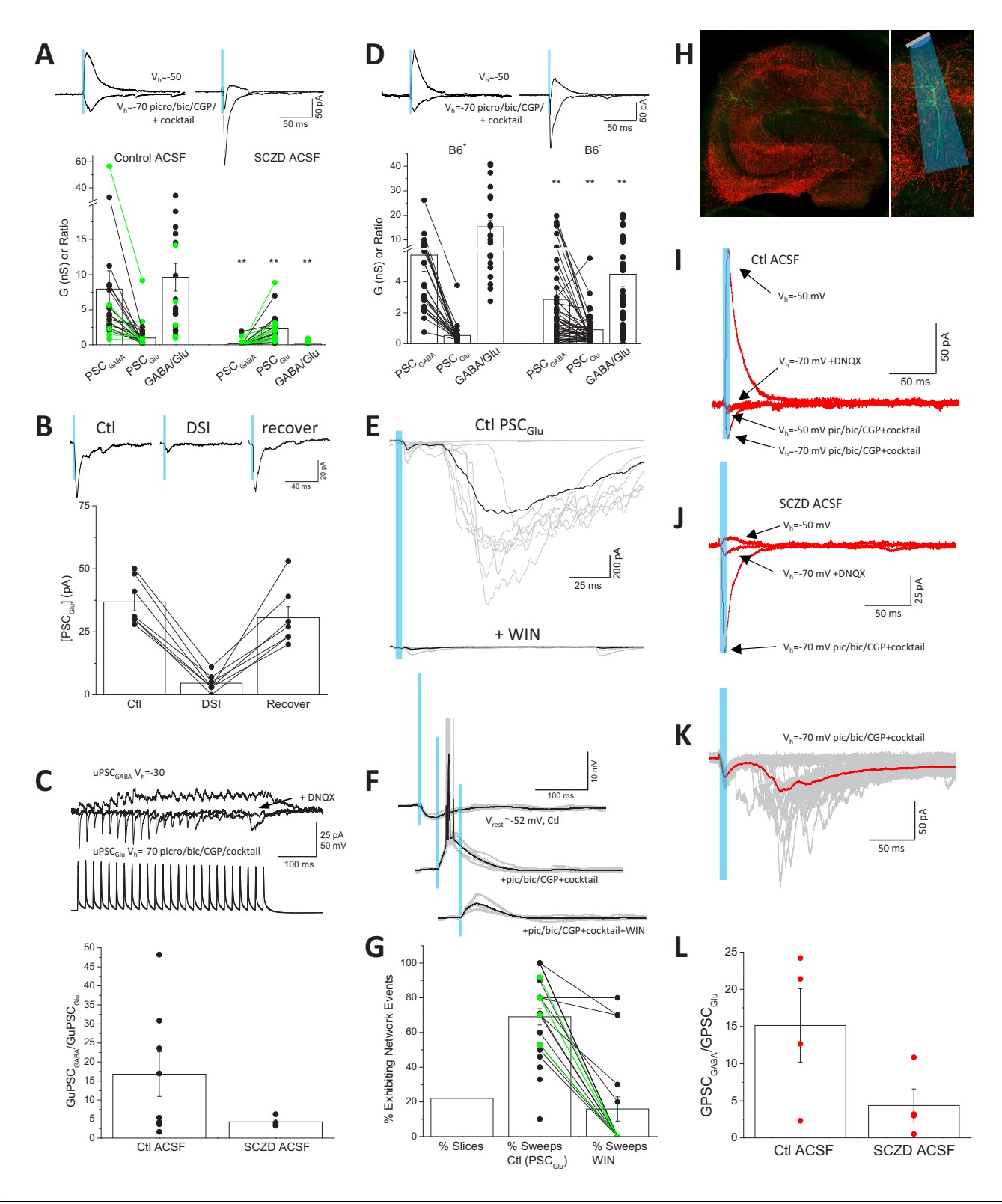

**Figure 5.** Enhanced glutamate release from CCK[+]VGluT3[+]INTs following GAD inhibition. (**A**) Traces from representative recordings (above) and group data summary (below) comparing light-evoked GABA and glutamate synaptic conductances observed in CA1 PCs of VGluT3-Cre:Ai32 or VGluT3-Cre: Chronos slices maintained in control (left, n = 22 cells from four mice) or SCZD supplemented ACSF (right, n = 23 cells from four mice; **p<0.01 vs. Control ACSF). (**B**) Traces from a representative recording (above) and group data plot (below) illustrating DSI of PSC_{Glu} evaluated in a subset SCZD

*Figure 5 continued on next page*

*Figure 5 continued*

treated slices (n = 7 cells from two mice). (C) Traces from a representative VGluT3⁺BC-CA1PC pair recording (above) and group data summary (below) illustrating enhanced glutamatergic, relative to GABAergic, transmission in slices maintained in SCZD (n = 4 pairs from two mice) compared to slices maintained in control ACSF (n = 8 pairs from six mice). Traces are averages of 3–5 sweeps per condition obtained by presynaptic train stimulation. (D) Traces from representative recordings (above) and group data plots (below) comparing light evoked GABA and glutamate synaptic conductances in CA1PCs of VGluT3-Cre:Ai32 mice maintained on B6⁺ diet (left, n = 24 cells from four mice) or B6⁻ diet (right n = 56 cells from nine mice **p<0.01 vs. B6⁺). (E) Example recording with GABAergic transmission blocked that displayed large polysynaptic network barrages following light evoked monosynaptic glutamatergic events (upper traces) that could be eliminated by the CB1R agonist WIN-55212–2 (lower traces). Individual trials/sweeps are shown overlayed in gray with the averaged events shown in black. (F) Example recording of a VGluT3-Cre:Ai32 CA1PC held in current clamp illustrating the transition from light-evoked GABAergic inhibition (upper traces) to suprathreshold glutamatergic excitation (middle traces) that is suppressed below threshold by WIN-55212–2 (lower traces). (G) Group data summary plot of the proportion of slices exhibiting light-evoked polysynaptic network barrages similar to those illustrated in E as well as the percentage of trials/sweeps in such recordings displaying network events in the absence (n = 25 slices from 11 mice) or presence of WIN-55212–2 (n = 17 slices, from six mice). (H) Images of a representative hippocampal section from a VGluT3-Cre mouse infected with AAV-mDlx-Flex-ChR2-mCherry (VGluT3-Cre:mDlxChR2) with a filled PC. (I–J) Averaged traces from representative PC recordings in slices from VGluT3-Cre:mDlxChR2 mice illustrating light evoked GABA and glutamate cotransmission under control (I) or SCZD (J) conditions. (K) Traces from a sample VGluT3-Cre:mDlxChR2 recording that exhibited light evoked polysynaptic network barrages. (L) Group data summary comparing PSC_GABA and PSC_Glu conductance ratios for VGluT3-Cre:mDlxChR2 slices maintained in control (n = 4 cells from one mice) or SCZD ACSF (n = 4 cells from one mice). Throughout the figure summary plots provide individual observations as well as group means ± SEM.

The online version of this article includes the following source data and figure supplement(s) for figure 5:

**Source data 1.** Data plotted in *Figure 5*.
**Figure supplement 1.** Supression of network activity by GABAergic output from CCK⁺VGluT3⁺ INTs.
**Figure supplement 1—source data 1.** Data plotted in *Figure 5—figure supplement 1*.

molecular marker for a subset of cortical CCK⁺ interneurons (*Fremeau et al., 2002*; *Somogyi et al., 2004*). However, no prior study has provided direct evidence for VGluT3-mediated vesicular synergy of GABA filling or glutamate cotransmission within CCK⁺VGluT3⁺INTS. A recent study observed reduced miniature IPSC (mIPSC) amplitudes following genetic ablation of VGluT3 generally or selectively within interneurons and interpreted the findings to reflect reduced SV GABA filling within CCK⁺VGluT3⁺INTS (*Fasano et al., 2017*). While these results indirectly support a role for VGluT3 in driving vesicular synergy, mIPSCs arise from multiple cohorts of hippocampal interneurons making it difficult to conclude that loss of VGluT3 selectively reduced the quantal GABA content of CCK⁺-VGluT3⁺INTS. In the same study, loss of VGluT3 (both generally and specifically in interneurons) increased mIPSC frequency and enhanced IPSCs evoked with bulk electrical stimulation (*Fasano et al., 2017*). These findings were interpreted to reflect a loss of glutamatergic tone at pre-synaptically expressed mGluRs controlling GABA release homo- or heterosynaptically. Indeed, mGluR4 antagonism enhanced evoked IPSCs (and mIPSC frequency) in wild type but not VGluT3 knockout mice indirectly implicating glutamate release from CCK⁺VGluT3⁺INTS in regulating GABA release.

Here, we directly confirm GABA and glutamate cotransmission from CCK⁺VGluT3⁺INTS through complementary optogenetic and synaptically coupled paired recording approaches. Although we did not specifically evaluate vesicular synergy, we note that CCK⁺VGluT3⁺INT uIPSC amplitudes were not obviously larger than that previously reported for the general CCK⁺ interneuron population (*Daw et al., 2009*; *Kohus et al., 2016*; *Lee et al., 2010*; *Vargish et al., 2017*; *Wyeth et al., 2017*; see also *Rovira-Esteban et al., 2017*; *Zimmermann et al., 2015*). Importantly, we found that VGluT3 expression in CCK⁺VGluT3⁺INTS exhibits a delayed developmental onset in contrast with neuronal populations that transiently express VGluT3 early in development during synaptogenesis or refinement (*Noh et al., 2010*; *Peirs et al., 2015*). The late onset of VGluT3 expression in cortical interneurons, combined with difficulty in selectively targeting VGluT3⁺ subsets, may partially explain the lack of prior evidence for glutamate release from any CCK⁺INTS as synaptic interrogation is often performed in acute slices from juvenile animals. Moreover, under basal conditions, the gluta-matergic component of CCK⁺VGluT3⁺INT-mediated transmission is absolutely dwarfed by the GABAergic component. We note that if GABAergic transmission is evaluated with stacked intracellu-lar chloride, as is common, GABAR antagonism would block >90% of a mixed PSC. With driving forces of opposite polarity, the strong GABAergic component fully shunted the glutamatergic com-ponent. Thus, selective targeting of CCK⁺VGluT3⁺INTS in adult mice, combined with systematic interrogation of their synaptic output is necessary to observe their glutamatergic phenotype.

We report that CCK⁺VGluT3⁺INTS exhibit a hitherto unexpected anatomical diversity comprising subsets of all known dendrite targeting CCK⁺ interneuron populations in addition to the expected BCs. We estimate that CCK⁺VGluT3⁺INTS account for only 7% of hippocampal interneurons (roughly half of all CCK⁺ interneurons, current findings and see *Bezaire and Soltesz, 2013*), yet this cohort exhibits profound influence over network excitability. Under basal conditions, the powerful GABAergic inhibition provided by CCK⁺VGluT3⁺INTS can be harnessed to dramatically dampen circuit excitability as evidenced by complete suppression of gamma oscillations (*Figure 5—figure supplement 1*). However, following amplification of their glutamatergic phenotype CCK⁺VGluT3⁺INTS can promote paradoxical network hyperexcitability which may be relevant to disorders associated with GAD dysfunction such as schizophrenia, depression, and Huntington's Disease as well as nutritional deficiencies in the co-enzyme pyridoxine (vitamin B6) (*de Jonge et al., 2017*; *Gupta et al., 2001*; *Hsu et al., 2018*; *Rush et al., 2012*; *Torrey et al., 2005*).

# Materials and methods

## Animals

All experiments were conducted in accordance with animal protocols approved by the National Institutes of Health (ASP# 17–045). VGluT3⁺ cells were genetically accessed and targeted using Tg (*Slc17a8-icre*)[1Edw/SealJ] mice (referred as VGluT3-Cre mice, *Grimes et al., 2011*), The Jackson Laboratory, stock #018147). Medial ganglionic eminence derived interneurons were targeted using Tg (*Nkx2-1-cre*)[2Sand/J] (referred to as Nkx2.1-Cre mice, The Jackson Laboratory, stock #008661). Male and female offspring from crosses with Cre-activated reporter lines conditionally expressing either tdTomato (Ai14 mice, The Jackson Laboratory, stock #007908) or ChR2-YFP (Ai32, The Jackson Laboratory, stock #012569) were used from P5 to 1 year as indicated. In addition, we used C57BL/6J wild-type mice for ISH and IHC studies that did not require transgenic reporting. Mice were housed and bred in a conventional vivarium with standard laboratory chow (or Teklad TD.01270 Pyridoxine Deficient Diet (B6⁻,~0.1 ppm) or Teklad TD. TD.01271 control diet with Pyridoxine HCl added back in (B6⁺,~10 ppm) where indicated, Envigo, Madison, WI) and water in standard animal cages under a 12 hr circadian cycle.

## Stereotaxic injections and viral vectors

For injections of viral vectors mice were anesthetized with 5% isoflurane and mounted in a stereotax (David Kopf Instruments Model 1900, Tujunga, CA). Topical lidocaine/prilocaine cream (2.5%/2.5%) and buprenorphine (0.1 mg/kg via subcutaneous injection) were provided for post-operative analgesia. Mice were provided with topical lidocaine/antibiotic ointment and ketoprofen daily for at least 3 days following surgery. VGluT3-Cre mice were injected with AAV-hSyn-FLEX-Chronos-GFP (UNC Vector Core), AAV-mDlx-FLEX-ChR2-mCherry, or AAV-CAG-FLEX-GFP (UNC Vector Core) into dorsal or mid ventral hippocampus via a glass micropipette attached to a syringe (Hamilton Company Inc, Reno, NV) and back filled with light mineral oil. Dorsal hippocampus was targeted using the following coordinates: 2.0 mm caudal and 1.5 mm lateral to bregma, and 1.1 mm deep from the dura. Mid-ventral hippocampus was targeted using the following coordinates: 2.8 mm caudal and 3.25 mm lateral to bregma, and 3.0 mm deep from the dura. At both injection sites, 350–600 nl of viral vector were injected at 100 nl/min following which the pipette was left in place for 5–10 min before removal. Mice recovered for at least 3 weeks before being used for recording or IHC.

To generate the AAV-mDlx-FLEX-ChR2-mCherry construct, the Addgene plasmid #83898 was digested with the enzymes SpeI and BsrGI, followed by the Gibson cloning of a reverse-complement of the ChR2-mcherry derived from the same plasmid and complemented with two incompatible lox sites (LoxP/Lox2272). The AAV was produced using standard procedures with the serotype one and the titer of the batch was estimated at 10E+12 viral particles per milliliter by qPCR.

## Antibodies

Primary antibodies included: Rat anti-SST (1:1000, MilliporeSigma, Cat# MAB354), Guinea Pig anti-CB1R (1:1000, Frontier Institute Co. Ltd., Cat# CB1-GP-Af530), Rabbit anti-VGluT3 (1:1000, Synaptic Systems, Cat# 131 004), Guinea Pig anti-VGAT (1:1000, Synaptic Systems, Cat# 131 006), Mouse anti-GAD1 (1:1000, Millipore, Cat# MAB5406), Mouse anti-Syt2 (1:1000, Zebrafish International

Resource Center, Cat# znp-1), Rabbit anti-CCK (1:1000, Frontier Institute Co. Ltd., Cat# CCK-pro-Rb-Af350), Mouse anti-PV (1:1000, Sigma-Aldrich, Cat# P3088), Rabbit anti-VIP (1:1000, ImmunoStar, Cat# 20077), Mouse anti-CRE (1:1000, Covance, Cat# MMS-106P-200), rabbit anti-VMAT2 (1:2000, Frontier Institute Co. Ltd., Cat# VMAT2-Rb-Af720), and guinea pig anti-panAMPAR: 1:1000; Frontier Institute Co. Ltd., Cat# panAMPAR-GP-Af580). Secondary antibodies were conjugated with Alexa Fluor dyes 488, 555, or 647 (1:1000; Thermo Fisher Scientific).

## IHC on perfused tissue

Mice were deeply anesthetized and tissue was fixed *via* transcardial perfusion with 30 mL of phosphate buffered saline (PBS) followed by 50 mL of 4% paraformaldehyde (PFA) in 0.1 M phosphate buffer (PB, pH 7.6). Brains were post-fixed overnight at 4°C when processed for immunostaining for SST, CB1R, VGluT3, VGAT, GAD1, Syt2, PV, and VMAT2 but not for more than 1 hr when preparing for VIP, CCK and CRE. Brains were cryopreserved in 30% sucrose and sectioned on a freezing microtome at 50 µm. Sections were rinsed in PB, blocked for 2 hr in 10% normal goat serum with 0.5% Triton X-100, and then incubated in primary antibody for 2 hr at room temperature or overnight at 4°C. Sections were then rinsed with PB and incubated in secondary antibodies (1:1000) and DAPI (1:2000) for 2 hr at room temperature. All antibodies were diluted in carrier solution consisting of PB with 1% BSA, 1% normal goat serum, and 0.5% Triton X-100. Sections were then rinsed, mounted on Superfrost glass slides, and coverslipped using Mowiol mounting medium and 1.5 mm cover glasses.

## Image acquisition and analysis

Confocal images were taken using a Zeiss 780 confocal microscope. For all slices with immunostained or genetically reported somatic signal, 50 µm thin sections were imaged using Plan-Apochromat 20x/0.8 M27 objective (imaging settings: frame size 1024 × 1024, pinhole 1AU, bit depth eight bit, speed 7, averaging 2). Confocal stacks were stitched using Zen Blue software (Zeiss) before importing them into Imaris software (Bitplane, version 9.2). Cell bodies were marked in Imaris software using the '*Spots*' function. VGluT3-Cre:Ai14 RFP$^+$ and PV$^+$ cell bodies were detected using the automatic function, with a signal detection radius of 10 µm. The Imaris '*Quality*' filter was set above an empirically determined threshold to maximize the number of detected cells while minimizing observed false-positives. CCK$^+$, SST$^+$, VIP$^+$ cell bodies were marked manually using the Imaris '*Spots*' function. ROI 3D borders, including CA1-CA2/3 (*Figure 1C*) and SO-SP-SR (*Figure 1D*), were drawn manually using the Imaris function '*Surfaces*'. Spots were then split within each ROI using the Imaris function '*Split Spots*'. Overlap of RFP$^+$ cells with other markers (PV, CCK, SST, VIP) was addressed by filtering the RFP$^+$ Spots above an empirically determined threshold intensity in the channel relative to the marker of interest. Each image with an automatic analysis by Imaris was checked by an expert and incorrectly identified cell bodies where refined if required. Data were imported and analyzed in Matlab (MathWorks).

To analyze VGluT3 terminal density throughout development, high-resolution single confocal images were taken using a 63x oil immersion objective on a Zeiss LSM 510 microscope (Zeiss Microscopy, Thornwood, NY). Counting was performed on four hippocampal sections from each animal. Quantitative analysis of VGluT3$^+$ puncta density in CA1/CA3 pyramidal cell (PC) layer was performed using ImageJ (NIH, Bethesda, MD, USA). Watershed parameters that best approximated manual puncta determination were established for each set of mice, and applied universally to all sections in that set. Images were converted to binary using ImageJ's automated 'Make Binary' function.

For high-resolution imaging to examine signal colocalization in putative axon terminals, confocal images were taken using a Zeiss 880-Airyscan microscope. 50 µm thin sections were imaged using Plan-Apochromat 63x/1.4 Oil DIC objective (imaging settings: airyscan optimal, zoom 3x). To consistently quantify the laminar position and colocalization of putative terminals, we imaged three coordinates-matched sections per mice and used n = 3 mice. Confocal stacks were centered on the CA1 pyramidal layer using the DAPI staining. Due to the high resolution of the images, terminals could be automatically detected using the '*Surfaces*' function in Imaris (detection radius of 10 µm, Imaris automatic detections settings were set above an empirically determined threshold for each channel and kept fixed for all animals). Overlap of VGluT3$^+$ terminals with other markers (CB1R, GAD1, VGAT, Syt2) was addressed by filtering the VGluT3$^+$ Spots above an empirically determined threshold intensity in the channel relative to the marker of interest. To address the homogeneity of the

terminals along the same parent axon, we traced continuous CB1R$^+$ axon segments using the Imaris function '*Filaments*' in '*manual mode*'. Only continuous segments longer than 40 μm, with a minimum of 5 VGluT3$^+$ release sites per examined segment were kept for further analysis. CB1R$^+$/VGluT3$^+$ and CB1R$^+$/VGluT3$^-$ terminals within each traced segment were marked manually using the Imaris '*Spots*' function. Data were imported and analyzed in Matlab (MathWorks).

### In situ hybridization

Fluorescent in situ hybridization was conducted according to RNAscope instructions for the HybEZ Hybridization System as reported in *Pelkey et al. (2015)*. Mice were deeply anesthetized with isofluorane, their brains flash frozen in liquid nitrogen and sectioned (10 μm) onto slides stored at −80 °C. For hybridization, sections were fixed in 4% PFA for 15 min at 4 ℃, dehydrated in an ascending ethanol series, pretreated with a protease using a kit for fresh frozen tissue (Advanced Cell Diagnostics Inc, No. 320842) and processed using the Fluorescent Multiplex Kit (Advanced Cell Diagnostics Inc, No. 320850) for probe hybridization, amplification and double fluorescent labeling (with dyes that excite at 488 and 550 nm). Probes were commissioned from RNAscope for *Sst* and *Slc17a8*. Sections were cover-slipped for examination on a Zeiss LSM710 confocal microscope.

### Slice preparation

Mice were anesthetized with isoflurane and then decapitated. The brain was dissected out in ice-cold sucrose substituted artificial cerebrospinal fluid (SSaCSF) containing the following (in mM): 90 sucrose, 80 NaCl, 3.5 KCl, 24 NaHCO$_3$, 1.25 NaH$_2$PO$_4$, 4.5 MgCl$_2$, 0.5 CaCl$_2$, and 10 glucose, saturated with 95% O$_2$ and 5% CO$_2$. Transverse dorsal or mid-ventral hippocampal slices (300–350 μm, in coronal or horizontal sections respectively) were cut using a VT-1200S vibratome (Leica Microsystems) and incubated submerged in the above solution at 32–34℃ for 30–40 min and then maintained at room temperature until use, either in the above solution or in recording aCSF consisting of the following (in mM): 130 NaCl, 3.5 KCl, 24 NaHCO$_3$, 1.25 NaH$_2$PO$_4$, 1.5 MgCl$_2$, 2.5 CaCl$_2$, and 10 glucose, saturated with 95% O$_2$ and 5% CO$_2$. Where indicated SSaCSF and aCSF for incubation and recording were supplemented with 4 mM semicarbazide (SCZD, Sigma) to inhibit GAD function. For pharmacological dissection of GABA and glutamate transmission aCSF was supplemented as indicated with (in μM): 50 picrotoxin, 10 bicuculline, 2 CGP 55845, 20 DNQX, 50 GYKI 53655, 1000 γ-DGG, 5 WIN 55212–2, 2 AM-251, 5 UBP 302, 10 LY 341495, and 100 cyclothiazide (CTZ) all from Tocris (UK). For gamma oscillation experiments following dissection in SSaCSF slices were transferred to an interface-style chamber (Warner Instruments, CT) containing humidified carbogen gas and perfused (1–1.5 ml/min, 34℃) with aCSF containing 2 mM each of MgCl$_2$, CaCl$_2$. For all electrophysiological investigations slices were incubated for at least 1 hr before recording.

### Slice electrophysiology

For patch-clamp recordings following recovery slices were transferred to an upright microscope (Zeiss Axioskop), perfused with aCSF (with or without SCZD as indicated) at 2–3 ml/min at a temperature of 32–34℃. Individual cells ells were visualized using a 40x objective using fluorescence and IR-DIC video microscopy. Electrodes were pulled from borosilicate glass (World Precision Instruments) to a resistance of 3–5 MΩ using a vertical pipette puller (Narishige, PP-830). Whole-cell patch-clamp recordings were made using a Multiclamp 700A amplifier (Molecular Devices), and signals were digitized at 20 kHz (Digidata 1322A, filtered at 3 kHz) for collection on a PC computer equipped with pClamp 9.2 or 10.4 software (Molecular Devices). Uncompensated series resistance ranged from 10 to 20 MΩ and was monitored continuously throughout recordings with −5 mV voltage steps. For current-clamp recordings of VGluT3-Cre:Ai14 RFP$^+$ interneurons and CA1PCs membrane potential was biased between −55 and −60 mV with internal solution containing (in mM): 130 K-gluconate, 5 KCl, 10 HEPES, 3 MgCl$_2$, 2 Na$_2$ATP, 0.3 NaGTP,, 0.6 EGTA, and 0.2% biocytin (calculated chloride reversal potential ($E_{Cl-}$) of −67 mV). For voltage-clamp recordings of CA1PCs two different internal solutions were used containing (in mM): A) 130 CsCl, 5 NaCl, 10 HEPES, 3 MgCl$_2$, 2 Na$_2$ATP, 0.3 NaGTP, 0.6 EGTA and 2 QX-314 with or without biocytin (0.2%) ($E_{Cl-}$~0 mV); and B) 130 Cs-methaneSO$_4$, 5 CsCl, 10 HEPES, 3 MgCl$_2$, 2 Na$_2$ATP, 0.3 NaGTP, 0.6 EGTA and 2 QX-314 with or without 0.2% biocytin ($E_{Cl-}$ ~ −62 mV). BAPTA (10 mM) was added to the intracellular solution where indicated.

To accurately determine neuronal resting membrane potential without disrupting the cell's intracellular environment, we monitored potassium-channel activation during depolarizing voltage ramps (from −100 to +200 mV) applied to cell-attached patches prior to breakthrough into the whole-cell configuration. After breaking into whole-cell configuration, membrane potential was biased to −55 to −60 mV by constant current injection. Input resistance (Rm) was measured using a linear regression of voltage deflections (±15 mV from resting potential,~60 mV) in response to 2 s current steps of six to ten different amplitudes (increment 5 pA). Membrane time constant ($\tau_m$) was calculated from the mean responses to 20 successive hyperpolarizing current pulses (−20 pA; 400 ms) and was determined by fitting voltage responses with a single exponential function. Action potential (AP) threshold was defined as the voltage at which the slope trajectory reaches 10 mV/ms. AP amplitude was defined as the difference in membrane potential between threshold and the peak. AP half-width was measured at the voltage corresponding to half of the AP amplitude. Afterhyperpolarization (AHP) amplitude was defined as the difference between action potential threshold and the most negative membrane potential attained during the AHP. These properties were measured for the first action potential elicited by a depolarizing 800 ms-long current pulse of amplitude just sufficient to bring the cell to threshold for AP generation. The adaptation ratio was defined as the ratio of the average of the last 2–3 interspike intervals relative to the first interspike interval during an 800-ms-long spike train elicited using twice the current injection necessary to obtain a just suprathreshold response. Firing frequency was calculated from the number of spikes observed during the same spike train. In some interneurons injection of hyperpolarizing current pulses induces pronounced 'sag' indicative of a hyperpolarization-activated cationic current ($I_h$) that activates following the initial peak hyperpolarization. To determine the sag index of each cell we used a series of 800 ms negative current steps to create V-I plots of the peak negative voltage deflection ($V_{hyp}$) and the steady state voltage deflection (average voltage over the last 200 ms of the current step; $V_{sag}$) and used the ratio of $V_{rest}-V_{sag}/V_{rest}-V_{hyp}$ for current injections corresponding to $V_{sag}=-80$ mV determined from polynomial fits of the V-I plots.

For optogenetic experiments in submerged slices, we applied brief blue light (470 nm, 2.5–5 ms) stimulation from an LED source (CoolLED PE4000, Andover, UK) through the 40x water immersion objective (~2.5 mW measured in the air at the level of the slice below the objective). Light evoked postsynaptic responses were acquired at 0.05–0.1 Hz and 5–20 individual responses in each condition were averaged to measure initial slopes, peaks, and decays as indicated. Given the divergent driving forces for GABAergic and glutamategic responses in our recordings, we converted peak amplitudes to conductances for direct comparison where indicated. CCK[+]VGluT3[+]INT-CA1PC paired recordings were performed as previously described (*Daw et al., 2010*; *Vargish et al., 2017*; *Wyeth et al., 2017*). Synaptic transmission was monitored by producing trains (25 pulses/50 Hz) or pairs (50 Hz) of action potentials in presynaptic CCK[+]VGluT3[+]INTS (held in current-clamp around −60 mV) every 10–30 s by giving 2 ms 1–2 nA current steps while holding the postsynaptic pyramidal cell at −30 to −70 mV as indicated in voltage-clamp mode. Presynaptic trains to probe for asynchronous release consisted of 25 presynaptic action potentials at 50 Hz. Basal unitary event properties for each cell were analyzed using 10–20 consecutive events obtained shortly after establishing the postsynaptic whole-cell configuration. Amplitudes reflect the average peak amplitude of all events including failures, potency is the average peak amplitude excluding failures. Paired pulse ratios (PPRs) for each cell were measured as the average amplitude of the second postsynaptic response divided by that of the first postsynaptic response for the paired responses obtained at 50 Hz. The coefficient of variation (CV) of uIPSCs was calculated as the SD of current amplitude divided by mean of the current amplitude. Asynchronous release was measured by deconvolution analysis as previously described (*Daw et al., 2009*). Briefly, an artificial miniature IPSC (mIPSC) was created using the rise time and decay kinetics (measured by single exponential fit) of a single unitary response and scaled to 20 pA. The postsynaptic trains were smoothed using 50 repetitions of binomial (Gaussian) smoothing. Fast fourier transforms (FFT) were then performed on the mIPSC and smoothed postsynaptic waveform. The FFT of the mIPSC was then divided, point-by-point, into the FFT of the postsynaptic waveform. The result of this division was then converted back into the time domain by inverse FFT, creating a release rate histogram. For a 50 Hz train, synchronous release was defined as the area under the curve (AUC) of the release rate histogram in the 5 ms bin immediately following the onset of the presynaptic current step, while asynchronous release was defined as the AUC in the 15 ms bin prior to the start of the next presynaptic current step. Synchronicity ratios

(SRs) were then calculated as synchronous release/asynchronous release for each presynaptic current step. To probe for DSI in both optogenetic experiments and paired recordings, CA1PCs cells were depolarized to 0 mV for 2–5 s to liberate endocannabinoids.

Local field potentials were recorded from slices maintained at interface with glass pipettes pulled from standard borosilicate glass (3–5 MΩ) and filled with aCSF placed in CA3 stratum pyramidale. Gamma oscillations were induced by applying 25 µM carbachol (Cch, Sigma-Aldrich, MO) as described (*Fisahn et al., 1998*). Blue or green widefield illumination was applied over the whole slice from approximately 1 cm above the interface level (~8 mW) for 10 s periods every 1–2 min after gamma oscillations were established with 10–20 min of Cch perfusion. Power spectra were generated from 10 s epochs before during and after the illumination to determine peak frequency, peak power, and gamma band power (20–80 Hz).

### Anatomical reconstructions

After biocytin filling during whole-cell recordings, slices were fixed with 4% paraformaldehyde and stored at 4°C then permeabilized with 0.3% Triton X-100 and incubated with Alexa Fluor 488 or Alexa Fluor 555-conjugated streptavidin. Resectioned slices were mounted on gelatin-coated slides using Mowiol mounting medium. Cells were visualized using epifluorescence microscopy (Olympus AX70) and images for representative examples were obtained with confocal microscopy (Leica TCS SP2 RS or Zeiss LSM 780).

All data were tested for normality and then we used either parametric or nonparametric statistical tests as appropriate.

## Acknowledgements

CJM is supported by the NICHD Intramural Research Program. GF is supported by NINDS grants NS081297, NS074972, and NIMH grants MH071679 and the Harvard's Dean Initiative. Both GF and JD are supported by R01-MH111529 and UG3MH120096, as well as support from the Simons Foundation. JD is supported by a gift from the Friends of FACES Foundation. Technical support provided by the NICHD Microscopy and Imaging Core.

## Additional information

### Funding

| Funder | Grant reference number | Author |
|---|---|---|
| *Eunice Kennedy Shriver* National Institute of Child Health and Human Development | Intramural research program | Chris J McBain |
| National Institute of Neurological Disorders and Stroke | NS081297 | Gordon Fishell |
| National Institute of Neurological Disorders and Stroke | NS074972 | Gordon Fishell |
| National Institute of Mental Health | MH071679 | Gordon Fishell |
| National Institutes of Health | R01-MH111529 | Jordane Dimidschstein Gordon Fishell |
| National Institutes of Health | UG3MH120096 | Jordane Dimidschstein Gordon Fishell |
| Simons Foundation | | Jordane Dimidschstein Gordon Fishell |
| Friends of FACES Foundation | | Jordane Dimidschstein |

The funders had no role in study design, data collection and interpretation, or the decision to submit the work for publication.

## Author contributions

Kenneth A Pelkey, Conceptualization, Data curation, Formal analysis, Supervision, Investigation, Methodology, Writing - original draft, Project administration, Writing - review and editing; Daniela Calvigioni, Xiaoqing Yuan, Data curation, Formal analysis, Investigation, Methodology, Writing - review and editing; Calvin Fang, Geoffrey Vargish, Connie Mackenzie-Gray Scott, Data curation, Formal analysis, Investigation, Writing - review and editing; Tyler Ekins, Kurt Auville, Data curation, Formal analysis, Investigation; Jason C Wester, Daniel Abebe, Investigation, Methodology; Mandy Lai, Formal analysis, Investigation, Methodology, Writing - review and editing; Steven Hunt, Investigation, Methodology, Writing - review and editing; Qing Xu, Resources, Methodology; Jordane Dimidschstein, Gordon Fishell, Resources, Methodology, Writing - review and editing; Ramesh Chittajallu, Conceptualization, Supervision, Writing - review and editing; Chris J McBain, Conceptualization, Supervision, Funding acquisition, Writing - review and editing

## Author ORCIDs

Kenneth A Pelkey (iD) https://orcid.org/0000-0002-9731-1336
Chris J McBain (iD) https://orcid.org/0000-0002-5909-0157

## Ethics

Animal experimentation: All experiments were conducted in accordance with animal protocols approved by the National Institutes of Health (ASP# 17-045).

## Decision letter and Author response

Decision letter https://doi.org/10.7554/eLife.51996.sa1
Author response https://doi.org/10.7554/eLife.51996.sa2

# Additional files

## Supplementary files

- Transparent reporting form

## Data availability

All data analyzed in this study are included in the manuscript/supporting files.

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
