## [Decision Letter]

**Acceptance summary:**

This manuscript by Pelkey and colleagues provides strong evidence that vesicular glutamate transporter type 3 (vGluT3) – immunoreactive hippocampal interneurons are capable of releasing glutamate. By using a combination of state-of-the-art approaches, the authors provide the first direct experimental evidence for glutamatergic transmission at these GABAergic output synapses. Compelling data shows that AMPA receptors on pyramidal neurons mediate the postsynaptic responses. The magnitude of glutamatergic postsynaptic currents is considerably smaller than that of the GABAergic component. However, after inhibition of GABA synthesis, the glutamatergic component increased and could lead to paradoxical excitation of the network. Most importantly, this manuscript also gives us the first insights about the physiological and pathophysiological significance of the glutamatergic component.

**Decision letter after peer review:**

Thank you for sending your article entitled "Paradoxical network excitation by glutamate release from VGluT3+ GABAergic interneurons" for peer review at *eLife*. Your article was evaluated by three peer reviewers, one of whom is a member of our Board of Reviewing Editors, and Gary Westbrook as the Senior Editor.

Major comments:

All three reviewers were impressed by the methodological broadness of the study and the strong experimental evidence supporting the conclusions with few exceptions, which are listed below.

1A) Two of the reviewers (#2 and #3) ask for additional data to support the observation that the expression of vGluT3 is developmentally delayed. Indeed, both reviewers found that the evidence to support this strong statement appears to be weak. The authors show an increase in the density of vGluT3-immunoreactive puncta (Figure 1A; note: no statistical analysis is reported), which can be explained by either the increased expression of vGlut3 in the boutons or the increased number of boutons. The authors should clarify which one it is by analyzing their existing data, e.g., by determining if the ratio of cb1+/vglut3+ and cb1+/vglut3- boutons changes during development. At the very least, some discussion of this point is warranted and/or the conclusions should be toned down. Moreover, measuring RFP signal intensity in cell bodies and axon terminals could reveal if vGluT3 levels steadily increase in these cells during development, or, alternatively whether there is an abrupt switch and as soon as a cell starts to express vGluT3, its levels reaches its maximum.

B) Along these lines, based on vGluT3-immunostaining and on the expression of a reporter driven by the promoter, the authors claim the delayed postnatal appearance of vGluT3. However, the delayed postnatal maturation of this interneuron type and their axonal arbor may also explain the experimental findings. Figure 2A can be interpreted by the arrival of additional vGluT3-positive interneurons to the hippocampal formation by P15 and Figure 1A can indicate intense axonal sprouting of these interneurons by P30. The authors could provide more direct evidence for their conclusion. Because the development of this specific cell population has not been studied in details, previous studies that investigated larger mixed interneuron populations may have overlooked a specific maturation delay in these cells. If all GABAergic cells (as assumed) are present already at P5 then the ratio of RFP-containing GAD-immunopositive interneurons should increase during postnatal maturation. Moreover, measuring RFP signal intensity in cell bodies and axon terminals could reveal if vGluT3 levels steadily increase in these cells during development, or, alternatively there is an abrupt switch and as soon as a cell starts to express vGluT3, its levels reaches its maximum.

C) Please discuss the seemingly contradictory result that cell numbers appear to increase then decrease with development (Figure 2A) while bouton densities appear to increase monotonically (Figure 1A).

2) Please provide neurolucida reconstruction of a few individual vGluT3-positive cells filled at different developmental stages to reveal if there is a proliferation of their axonal arbor, which, may explain the data presented in Figure 1A.

3) Optogenetic driving of mixed inputs revealed in only 17 out of 53 synaptically coupled vGluT3 interneuron – pyramidal cell pair an AMPA component. This is surprising given that AMPA is present everywhere in the PC somata (Figure 4—figure supplement 1A) and vGlut3 is present in every bouton of the RFP cells (Figure 1C). What could be the potential explanation? Is it possible that the somatic glutamatergic component is only present at basket cells, but not at dendritically targeting cells (or perhaps the other way around)? Did the authors perform morphological reconstruction of the presynaptic interneurons that did or did not elicit an AMPA response? What would be the effect of glutamate corelease at dendrites (by Schaffer collateral-associated) or at distal tufts (by perforant path-associated cells)?

4) Please provide larger font sizes to all figures, particularly Figure 1B, 1C, 1D; 2B, 3B and E, 4A, C-K. The panel letters Figure 4E-F are not visible. Some of the x and y-axis labels are very small (Figure 1B; 2B).

---

## [Author Response]

Major comments:All three reviewers were impressed by the methodological broadness of the study and the strong experimental evidence supporting the conclusions with few exceptions, which are listed below.1A) Two of the reviewers (#2 and #3) ask for additional data to support the observation that the expression of vGluT3 is developmentally delayed. Indeed, both reviewers found that the evidence to support this strong statement appears to be weak. The authors show an increase in the density of vGluT3-immunoreactive puncta (Figure 1A; note: no statistical analysis is reported), which can be explained by either the increased expression of vGlut3 in the boutons or the increased number of boutons. The authors should clarify which one it is by analyzing their existing data, e.g., by determining if the ratio of cb1+/vglut3+ and cb1+/vglut3- boutons changes during development. At the very least, some discussion of this point is warranted and/or the conclusions should be toned down. Moreover, measuring RFP signal intensity in cell bodies and axon terminals could reveal if vGluT3 levels steadily increase in these cells during development, or, alternatively whether there is an abrupt switch and as soon as a cell starts to express vGluT3, its levels reaches its maximum.

As suggested, we performed developmental characterizations of VGluT3 expression in CA1 perisomatic innervating CB1R^+^ terminals of wild type mice and additionally in RFP^+^ terminals of VGluT3-Cre:Ai14 mice. We observed a high proportion of VGluT3 lacking RFP^+^ perisomatic terminals in young VGluT3-Cre:Ai14 mice, consistent with delayed VGluT3 accumulation in preexisting axons (incorporated as revised Figure 2—figure supplement 2A). Moreover, the proportion of VGluT3^+^ CB1R^+^ perisomatic terminals significantly increases from postnatal day 10 to 90 with no significant increase in overall CB1R^+^ terminal density consistent with delayed accumulation of VGluT3 in a subset of hippocampal resident CCK^+^BCs (incorporated as Figure 2—figure supplement 2B). Finally, anatomical evaluation of individual VGluT3-Cre:Ai14 reported neurons over development, as requested by the reviewers, revealed mature axon profiles as early as P13 supporting the conclusion of delayed VGluT3 expression within anatomically mature BCs rather than delayed axon extension following VGluT3 expression (incorporated as revised Figure 2—figure supplement 2C). All of these new datasets are referred to subsection “CCK^+^VGluT3^+^205 INTS exhibit rich anatomical diversity and typical GABAergic transmission properties” of the revised manuscript text.

Note that the revision experiments afforded a natural opportunity to extend our original VGluT3^+^ terminal density developmental characterization. Accordingly, the data have been incorporated into revised Figure 1A along with statistical comparisons the reviewers indicated were lacking in the original manuscript.

B) Along these lines, based on vGluT3-immunostaining and on the expression of a reporter driven by the promoter, the authors claim the delayed postnatal appearance of vGluT3. However, the delayed postnatal maturation of this interneuron type and their axonal arbor may also explain the experimental findings. Figure 2A can be interpreted by the arrival of additional vGluT3-positive interneurons to the hippocampal formation by P15 and Figure 1A can indicate intense axonal sprouting of these interneurons by P30. The authors could provide more direct evidence for their conclusion. Because the development of this specific cell population has not been studied in details, previous studies that investigated larger mixed interneuron populations may have overlooked a specific maturation delay in these cells. If all GABAergic cells (as assumed) are present already at P5 then the ratio of RFP-containing GAD-immunopositive interneurons should increase during postnatal maturation. Moreover, measuring RFP signal intensity in cell bodies and axon terminals could reveal if vGluT3 levels steadily increase in these cells during development, or, alternatively there is an abrupt switch and as soon as a cell starts to express vGluT3, its levels reaches its maximum.

Based on the IHC and anatomical revision data outlined in response to point 1A above the most parsimonious explanation for increasing VGluT3^+^ terminal densities (and RFP^+^ cell densities in VGluT3-Cre:Ai14 mice) is a delayed onset of VGluT3 promoter utilization and protein expression in a subset of interneurons already resident in the hippocampus within the first postnatal week. Importantly, such delayed accumulation of a cell type specific protein during development is entirely analogous to the widely accepted delayed upregulation of parvalbumin within parvalbumin basket cells, and also CAMKII within pyramidal cells (as opposed to late postnatal hippocampal infiltration or process extension by parvalbumin expressing interneurons or CAMKII expressing pyramidal cells). While we did not compare “RFP signal intensity” in cell bodies versus terminals, our new findings clearly illustrate that VGluT3 signal lags terminal RFP and CB1R expression as well as axon arborization (revised Figure 2—figure supplement 2A-C). It is important to note that the RFP signal is simply a proxy for promoter utilization and not a direct indicator of VGluT3 protein levels or subcellular localization. Our observations of RFP^+^VGluT3^-^ perisomatic terminals early in development is consistent with axon extension prior to saturating “maximum” VGluT3 levels at the time of axon extension. Similarly, our characterization of CB1R/VGluT3 colocalization indicates that VGluT3 developmentally accumulates in preexisting perisomatic terminals as the overall density of CB1R^+^ terminals did not increase.

Note that the revision experiments afforded a natural opportunity to extend our original VGluT3-Cre:Ai14 RFP^+^ cell density developmental characterization. Accordingly, the data have been incorporated into revised Figure 2A along with statistical comparisons.

C) Please discuss the seemingly contradictory result that cell numbers appear to increase then decrease with development (Figure 2A) while bouton densities appear to increase monotonically (Figure 1A).

As stated above our revision afforded an opportunity to extend the original observations in Figures 1A and 2A describing the developmental profiles of native VGluT3^+^ terminal density and VGluT3-Cre:Ai14 RFP^+^ cell density. Statistical evaluations of the revised data sets indicate that VGluT3^+^ terminal density plateaus beyond 4 weeks of age while RFP cell density plateaus beyond 2 weeks of age. Importantly, RFP signal within VGluT3-Cre:Ai14 mice is simply a proxy for VGluT3 promoter utilization and not a direct indicator of VGluT3 protein levels, trafficking, or subcellular localization. Moreover, in our analyses RFP somatic signal is digitally scored as either present or absent with no consideration of abundance. Thus, temporal differences in achieving plateau levels likely reflect differences in manufacturing, trafficking, and targeting of VGluT3 vs RFP. Indeed, under our conditions RFP^+^ cell reporting would be expected to reach a maximum shortly after initial promoter availability while VGluT3 terminal accumulation would lag due to trafficking and targeting requirements. The modest (not significant) decrease in RFP^+^ cell density beyond P15 likely reflects hippocampal volume dilution of the sparse cell somas located throughout the expanding neuropil.

2) Please provide neurolucida reconstruction of a few individual vGluT3-positive cells filled at different developmental stages to reveal if there is a proliferation of their axonal arbor, which, may explain the data presented in Figure 1A.

The reconstructions are provided in revised Figure 2—figure supplement 2C. Anatomical evaluation revealed mature axon profiles as early as P13 supporting the conclusion of delayed VGluT3 expression within anatomically mature BCs rather than delayed axon extension following VGluT3 expression.

3) Optogenetic driving of mixed inputs revealed in only 17 out of 53 synaptically coupled vGluT3 interneuron – pyramidal cell pair an AMPA component. This is surprising given that AMPA is present everywhere in the PC somata (Figure 4—figure supplement 1A) and vGlut3 is present in every bouton of the RFP cells (Figure 1C). What could be the potential explanation? Is it possible that the somatic glutamatergic component is only present at basket cells, but not at dendritically targeting cells (or perhaps the other way around)? Did the authors perform morphological reconstruction of the presynaptic interneurons that did or did not elicit an AMPA response? What would be the effect of glutamate corelease at dendrites (by Schaffer collateral-associated) or at distal tufts (by perforant path-associated cells)?

To clarify, optogenetic activation of VGluT3 cells consistently revealed a glutamatergic component in all pyramidal cells evaluated. In paired recordings between VGluT3Cre-Ai14 reported presynaptic cells and postsynaptic pyramidal cells with a confirmed GABAergic component only 17/53 pairs exhibited a measurable glutamatergic component (Figure 4J-K). We do not know why a significant portion of such paired recording failed to exhibit a glutamatergic component and thus are hesitant to speculate in the manuscript. Anecdotally, we were able to observe glutamatergic events from both BCs and DTIs; however, we do not have sufficient anatomical data for meaningful analysis. Two major differences between the optogenetic versus paired recording approaches are contribute to the differential success:

A) The much greater number of release sites engaged by optogenetic stimulation will facilitate transmitter pooling and maximize the chances of engaging release sites with AMPARs in sufficient proximity to sense the released (maybe pooled) transmitter. Note that ubiquitous expression of somatic AMPARs by pyramids does not necessarily equate to discrete localization to postsynaptic sites innervated by each individual VGluT3^+^ axon.

B) Presynaptic cellular dialysis in paired recordings may reduce the chances to observe a glutamatergic component. Note that the experimental design demands that the presynaptic cell is dialyzed for approximately 10 minutes to confirm then block the GABAergic component before assaying for the glutamatergic event. Such cellular dialysis can disrupt unitary glutamatergic output even from pyramidal cells (Biro, Holderith and Nusser J. Neurosci., 2005).

4) Please provide larger font sizes to all figures, particularly Figure 1B, 1C, 1D; 2B, 3B and E, 4A, C-K. The panel letters Figure 4E-F are not visible. Some of the x and y-axis labels are very small (Figure 1B; 2B).

We have made the requested changes throughout the figures.